



# A uniform $p\mathrm{CO_2}$ climatology combining open and coastal oceans

Peter Landschützer[1], Goulven G. Laruelle[2,], Alizee Roobaert[2], and Pierre Regnier[2]

[1]Max Planck Institute for Meteorology, Hamburg, Germany
[2]Department Geoscience, Environment & Society (DGES), Université Libre de Bruxelles, Brussels, CP16002, Belgium

*Correspondence to:* Peter Landschützer (peter.landschuetzer@mpimet.mpg.de)

**Abstract.**

In this study, we present the first combined open and coastal ocean $p\mathrm{CO_2}$ mapped monthly climatology (Landschützer et al. (2020), doi: 10.25921/qb25-f418, https://www.nodc.noaa.gov/ocads/oceans/MPI-ULB-SOM_FFN_clim.html) constructed from observations collected between 1998 and 2015 extracted from the Surface Ocean $\mathrm{CO_2}$ Atlas (SOCAT) database. We combine
two neural network-based $p\mathrm{CO_2}$ products, one from the open ocean and the other from the coastal ocean, and investigate their consistency along their common overlap areas. While the difference between open and coastal ocean estimates along the overlap area increases with latitude, it remains close to 0 $\mu$atm globally. Stronger discrepancies, however, exist on the regional level resulting in differences that exceed 10% of the climatological mean $p\mathrm{CO_2}$, or an order of magnitude larger than the uncertainty from state of the art measurements. This also illustrates the potential of such analysis to inform the measurement community
about the locations where additional measurements are essential to better represent the aquatic continuum and improve our understanding of the carbon exchange at the air water interface. A regional analysis further shows that the seasonal carbon dynamics at the coast-open interface are well represented in our climatology. While our combined product is only a first step towards a true representation of both the open ocean and the coastal ocean air-sea $\mathrm{CO_2}$ flux in marine carbon budgets, we show it is a feasible task and the present data product already constitutes a valuable tool to investigate and quantify the dynamics of
the air-sea $\mathrm{CO_2}$ exchange consistently for oceanic regions regardless of its distance to the coast.

## 1 Introduction

Since the beginning of the industrial revolution, human activities such as fossils fuel energy combustion, cement production and land used change have emitted a large quantity of carbon dioxide ($\mathrm{CO_2}$) in the atmosphere disturbing the global carbon cycle
and inducing global climate change (Friedlingstein et al., 2019). The ocean plays a fundamental role in understanding the fate of anthropogenic carbon dioxide since it acts as a $\mathrm{CO_2}$ sink and removes roughly 25 % of the anthropogenic $\mathrm{CO_2}$ emitted into the atmosphere every year (Friedlingstein et al., 2019). However, uncertainties are still associated to this estimate, especially in highly heterogeneous and/or poorly monitored regions such as the Arctic Ocean, the southeast Pacific and the coastal ocean (Regnier et al., 2013; Laruelle et al., 2014). Reducing the uncertainty of current marine $\mathrm{CO_2}$ sink estimates is however essential



to improve our understanding of the underlying processes controlling the contemporary and future distribution of anthropogenic $CO_2$ between atmosphere, land and ocean.

While current oceanic $CO_2$ sink estimates largely rely on the output from hindcast simulations of global biogeochemistry models (Sarmiento et al., 2010; Le Quéré et al., 2018) and atmospheric as well as oceanic inverse models (Mikaloff Fletcher
et al., 2006; Gruber et al., 2009; Wanninkhof et al., 2013), several observation-based estimates built on surface ocean $CO_2$ measurements emerged in the past years (Landschützer et al., 2014; Rödenbeck et al., 2015; Zscheischler et al., 2017; Laruelle et al., 2017). These estimates are, in part, the result of the community effort that led to the establishment of two large and still growing collections of surface ocean $CO_2$ measurements, namely the LDEO database (Takahashi et al., 2018) and the Surface Ocean $CO_2$ Atlas (SOCAT) database (Pfeil et al., 2013; Sabine et al., 2013; Bakker et al., 2014, 2016).

The oceanic uptake of $CO_2$ is directly proportional to the partial pressure difference of $CO_2$ ($\Delta pCO_2$) between the oceanic surface water and the atmosphere. Therefore, the increase in available observations from roughly 6 million in the first release of the SOCAT database (SOCATv1.5) in 2011 (Pfeil et al., 2013) to a total of more than 23 million observations gathered in version 6 (SOCATv6) (Bakker et al., 2016), resulted in increasingly detailed and accurate observational-based studies investigating the ocean carbon sink (Rödenbeck et al., 2015). While earlier work such as Takahashi et al. (2009) focused on the long term mean
$CO_2$ uptake and its spatial and seasonal variations, the sustained increase in data density now allows investigating temporal variations on longer time scales (Rödenbeck et al., 2014; Majkut et al., 2014; Landschützer et al., 2014; Rödenbeck et al., 2015; Jones et al., 2015; Landschützer et al., 2016), suggesting a variable ocean $CO_2$ sink on interannual to decadal timescales (Rödenbeck et al., 2015; Landschützer et al., 2015). These estimates, however, suffer from two main sources of uncertainty. The first related to the kinematic transfer of $CO_2$ across the air-sea interface (Wanninkhof and Trinanes, 2017; Roobaert et al.,
2018) and a second, less well quantified, source related to the interpolation of sparse surface ocean partial pressure of $CO_2$ data (e.g. Rödenbeck et al., 2015; Landschützer et al., 2014).

Similar to the open ocean, coastal regions - defined here following the broad SOCAT boundary definition of 400km distance from shore used in Laruelle et al. (2017) - are also recognized as a $CO_2$ sink for the atmosphere (e.g. Laruelle et al., 2014) but have long been constrained using scarce data of uneven spatial and temporal distribution (Thomas et al., 2004; Borges
et al., 2005; Cai et al., 2006; Chen and Borges, 2009; Laruelle et al., 2010; Cai, 2011; Chen et al., 2013; Dai et al., 2013). Therefore, because of the strong physical and biogeochemical heterogeneity of the coastal ocean, a proper representation of the spatio-temporal patterns in $CO_2$ fluxes could only be achieved in the best-monitored regions of the world (Laruelle et al., 2014). More recently, the application of neuronal network-based interpolation methods similar to those applied for the open ocean resulted in the first continuous global $pCO_2$ climatology for the coastal ocean, which improved the estimation of coastal
carbon sink and its spatial variability (Laruelle et al., 2017; Roobaert et al., 2019). It is also only very recently that studies have performed a global-scale analysis of the seasonal variability of the air-water $CO_2$ exchange (Roobaert et al., 2019).

As an additional challenge, many different boundaries have been used to delineate the frontier between coastal and open ocean waters in the past (Walsh, 1988; Borges et al., 2005; Liu et al., 2010; Laruelle et al., 2010, 2013). The choice of a specific delineation has nevertheless important implications for the quantification of the coastal $CO_2$ sink as well as the
adjacent open ocean sink and their temporal trends (Laruelle et al., 2014, 2018). Including the contribution of the coastal ocean





in observation-based air-sea $CO_2$ exchange estimates, i.e. the aim of this study, is not only important in order to improve the quantification of the present-day global ocean sink which has so far been based on open ocean data only, but also to properly analyse the trends and spatio-temporal variabilities of all ocean waters in a consistent manner. Several recent studies have indeed suggested that, as a whole, the intensity of the $CO_2$ sink per unit area could be stronger in coastal regions than in the

open ocean (Borges et al., 2005; Cai, 2011; Laruelle et al., 2010, 2014), whereas Roobaert et al. (2019) suggests that this difference stems from the uneven latitudinal distribution of surface areas between coastal and open ocean but that adjacent open and coastal regions behave similarly.

This distinct behavior of the coastal ocean, with possibly a stronger present-day uptake and a fast-increasing air-sea $pCO_2$ gradient on decadal timescales is not only relevant for today's quantification of the ocean sink, but also for constraining the

anthropogenic perturbation of the marine $CO_2$ sink. So far, the latter has only been estimated by assuming similar changes in open ocean and coastal seas $CO_2$ flux densities since pre-industrial times (Wanninkhof et al., 2013; Regnier et al., 2013) while other studies have proposed larger anthropogenic perturbations for the shallow parts of the ocean by mostly relying on conceptual modeling approaches (e.g. Bauer et al., 2013). The need for a unified coastal-open ocean $pCO_2$ climatology is further reinforced by the recent upward revision of the pre-industrial global ocean $CO_2$ outgassing fuelled by the river carbon

loop (Kwon et al., 2014; Resplandy et al., 2018). As a significant fraction of this $CO_2$ outgassing derived from terrestrial carbon inputs likely takes place near the coast or across the coastal-open ocean transition, it is important to establish a global ocean $pCO_2$ climatology that can be used as benchmark for increasingly refined models reconstructing the historical evolution of the marine carbon sink.

As a first step towards this goal, we combine two state-of-the-art sea surface observational $pCO_2$ products for the open

ocean and the coastal regions to create a common global $pCO_2$ climatology that covers the entirety of the global ocean to better represent the spatio-temporal patterns in the overall marine carbon sink. The combined data product is the first continuous coastal-open ocean $pCO_2$ climatology constructed with a near-uniformly treated dataset. It also includes the Arctic Ocean, which was not considered in previous open ocean global analyses (Landschützer et al., 2014; Landschützer et al., 2016) and was only partly included in the coastal $pCO_2$ climatology of Laruelle et al. (2017). In spite of its relatively limited surface area

and a significant proportion of seasonal sea ice coverage which prevents most of the gas exchange (Lovely et al., 2015), the Arctic Ocean and its extensive continental shelves is a major contributor of the global coastal $CO_2$ sink (Yasunaka et al., 2016), displaying some of the most intense air-water $CO_2$ exchange rate per unit area (Roobaert et al., 2019). The incorporation of these high-latitude regions is thus essential to avoid a bias when analyzing the role of the coastal zone on the global ocean $CO_2$ sink.

Here, using the new global ocean $pCO_2$ climatology as well as the individual coastal and open ocean data products, we investigate how well the coastal-open ocean continuum is reconstructed through statistical error analysis. In particular, our goal is to address the following research questions: 1) to what extent reconstructed $pCO_2$ estimates from both products agree with one another in regions where they overlap; 2) to what extent eventual mismatches are related to data sparsity, both for the temporal $pCO_2$ mean and the seasonal climatology.





## 2 Methods

### 2.1 Open ocean and coastal datasets

Our analysis is based on two recently published sea surface $pCO_2$ data products. The first one, updated from Landschützer et al. (2016), covering broadly the open ocean at a distance of $1°$ off the coast and, the second dataset, by Laruelle et al. (2017),

covering the coastal domain plus the adjacent open ocean up until 400km away from the shoreline for a total surface area of $70x10^6$ km2. Both datasets are based on the same neural network interpolation method, i.e. the SOM-FFN (Self Organizing Map - Feed Forward Neural Network) method (Landschützer et al., 2013). While the individual datasets (from here onward "NN$_{open}$" for the open ocean dataset and "NN$_{coast}$" for the coastal ocean dataset) have been extensively described and validated in their individual publications Landschützer et al. (2014); Landschützer et al. (2016); Laruelle et al. (2017), we present here a

short summary of each product including their most recent updates and the procedure used to merge both datasets.

  The SOM-FFN method consists of a 2-steps interpolation approach. First, a marine region (i.e. either open ocean or coastal ocean) is divided into biogeochemical provinces based on similarities within selected environmental $CO_2$ driver data. Secondly, the non-linear relationship between a second set of driver data and available sea surface $pCO_2$ data from the SOCAT database is established and can then be used to fill gaps where no observations exist (see Landschützer et al., 2013). Both open and

coastal ocean applications rely on satellite and reanalysis data, but different sets of environmental driver variables are used. For the open ocean analysis, sea surface temperature, salinity, mixed layer depth, chlorophyll-a and atmospheric $CO_2$ are used as proxy variables.

  While leaving NN$_{coast}$ unchanged to its original publication (Laruelle et al., 2017), we here provide two updates to NN$_{open}$ compared to its previous publications (see Landschützer et al., 2013, 2014). Firstly, we replaced the mixed layer depth proxy

of the NN$_{open}$ from de Boyer Montegut et al. (2004) to the Argo based MIMOC product (Schmidtko et al., 2013) as it allows us to expand our analysis region, creating a maximum overlap area between NN$_{open}$ with NN$_{coast}$, while the error statistics of the method remain nearly unchanged. Secondly, for completeness, we also include the Arctic Ocean in NN$_{open}$, allowing the comparison between products to be extended to the high latitudes. In order to achieve this, the Arctic Ocean was assigned its own stand-alone oceanic biome in the SOM procedure (see Landschützer et al., 2013). Previous global-scale studies avoided

the Arctic Ocean (Takahashi et al., 2009; Landschützer et al., 2014), however more recent studies by Yasunaka et al. (2016) illustrate that the increase in measurements makes a reconstruction feasible. Due to its uniqueness in its seawater properties, we find that assigning the Arctic Ocean a stand-alone biome, which is not varying in time, provides the best reconstruction. This way, the Arctic $pCO_2$ is only determined by Arctic Ocean measurements (starting at 79N in the Atlantic Ocean) while Arctic Ocean measurements are not influencing other biomes. Hence, the remainder of the global ocean remains unchanged by

this addition and the $pCO_2$ product is thus considered the same as the one presented in (Landschützer et al., 2016).

  The NN$_{open}$ and NN$_{coast}$ are all available at the same monthly temporal resolution but are applied at different spatial resolutions. While NN$_{open}$ uses a $1°x1°$ resolution, the coastal $pCO_2$ data product is constructed at a higher $0.25°x0.25°$ resolution to better capture the spatial heterogeneity of the coastal zone. Thus, in order to combine and compare the products at the same spatial resolution, we divided each $1°x1°$ grid cell of the open ocean into 16 equal $0.25°x0.25°$ bins . NN$_{coast}$

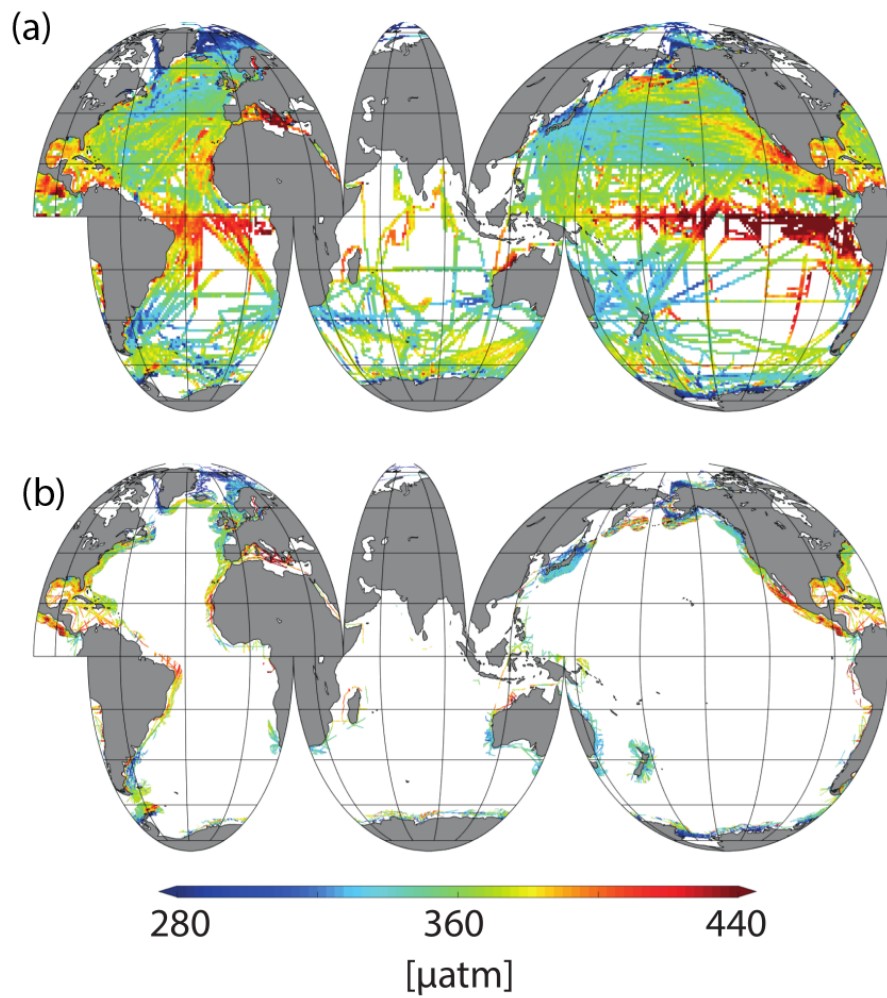

**Figure 1.** Gridded (a) $1° \times 1°$ open ocean and (b) $0.25° \times 0.25°$ coastal ocean $pCO_2$ data values extracted from the SOCATv5 database from 1998 through 2015. Each value on the maps represents the mean of all values available within each grid cell for the period considered.

combines observations from 1998 through 2015 using SOCATv4, whereas NN$_{open}$ uses SOCATv5 data from 1982 through 2016. In this study, we constructed a climatological mean for the common period covered by both products (1998-2015). Despite the use of different versions of the SOCAT database used to generate the two $pCO_2$ products (SOCATv4 vs SOCATv5) we expect little influence on our results, since most of the new data introduced into SOCATv5 compared to SOCATv4 were
5  added in the later years and, in particular, 2016 which is excluded from our analysis. Figure 1 illustrates the temporal mean of all available $pCO_2$ observations extracted from the SOCATv5 dataset for the 1998-2015 period.

Figure 2 shows the climatological mean $pCO_2$ for both NN$_{open}$ (Landschützer et al., 2016) and NN$_{coast}$ (Laruelle et al., 2017). The data products rely on sea masks that lead to a common overlap area at the coastal-open ocean transition of roughly

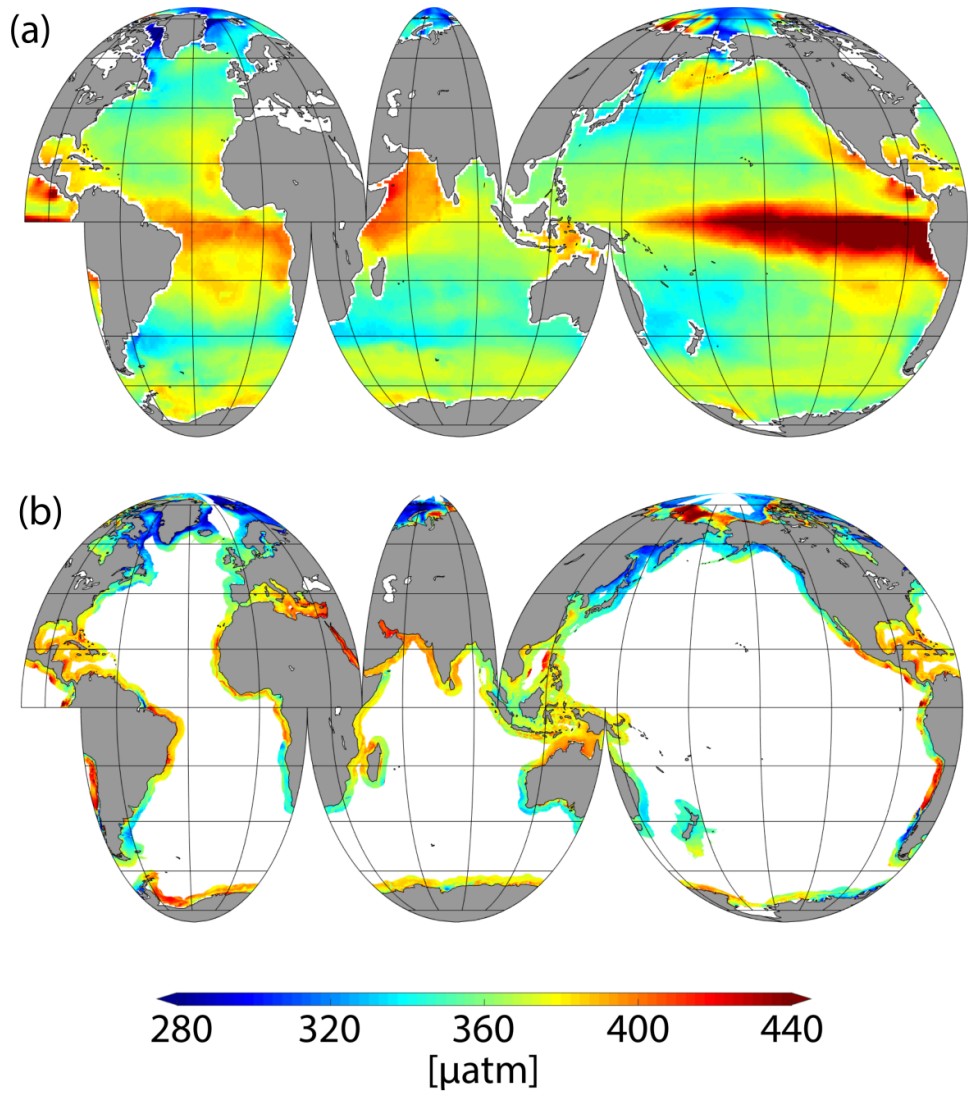

**Figure 2.** climatological mean of the (a) $1° \times 1°$ open ocean $p$CO$_2$ product by Landschützer et al. (2016) and (b) $0.25° \times 0.25°$ the coastal ocean $p$CO$_2$ product by Laruelle et al. (2017) for the 1998-2015 period

$42 \times 10^6$ km$^2$, reflecting the lack of a commonly recognized definition of the boundary between both environments. While the landward limit of the NN$_{open}$ is located on average at around $1°$ (or roughly 100km) off shore, NN$_{coast}$ extends from the coastline to either 400km offshore or the 1000 m isobath, whichever is encountered first. The bathymetry used follows the SOCAT coastal definition (Pfeil et al., 2013) and excludes estuaries and inner water bodies (Laruelle et al., 2013, 2017). This
5    overlap area is the subject of our error analysis described below.

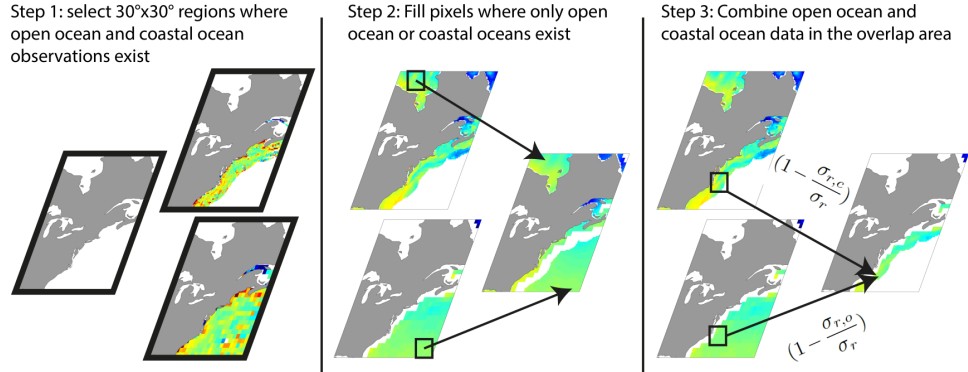

**Figure 3.** Schematic illustration of the merging steps. Step 1 shows an illustrative example of one 30°x30° box that includes both coastal and open ocean SOCAT observations. in Step 2 empty grid cells within the 30°x30° box are filled with coastal ocean as well as open ocean datapoints and in Step 3 open ocean and coastal ocean datapoints are combined where both exist.

## 2.2 Merging algorithm

The combination of the two data products takes place in three steps which are illustrated in Figure 3. In a first step, we divide the globe into a raster of coarse 30°x30° boxes starting at 90°N and 180°W. The large box size ensures that, even in remote regions, observations from both open ocean and coastal ocean are represented in the overlap area. We then investigate the overlap area for each raster box individually. In a second step, within each 30°x30° box, the pixels that are only covered by either $NN_{open}$ or $NN_{coast}$ are assigned their respective $pCO_2$ value. In a third step, all pixels where open ocean and coastal ocean $pCO_2$ products overlap, that is, all 0.25°x0.25° pixels with co-located $pCO_2$ values in the open ocean and coastal ocean datasets, are identified. To assign a $pCO_2$ value in this overlap area, we weight the open and coastal $pCO_2$ estimates by their standard error relative to the SOCATv5 open and SOCATv5 coastal ocean datasets, respectively. We calculate the standard error at the scale of each 30°x30° raster, as at this larger scale regions enough observations are available to provide an error statistic. To implement this scheme, we first calculate the standard error on each 30°x30° box as:

$$\sigma_i = \frac{RMSE_i}{\sqrt{N_i}} \tag{1}$$

where RMSE is the root mean square error of the open and coastal datasets with respect to the SOCATv5 gridded observations, N is the number of available gridded data from SOCATv5 available in a given 30°x30° raster box and the subscript i refers to either $NN_{open}$ or $NN_{coast}$, respectively. Since we have simply divided the open ocean from a 1°x1° grid into 16 equal 0.25°x0.25° bins, we use an effective number of $N^{eff}$=N/16 for the open ocean. We do not account for autocorrelation in our calculations since we are only interested in the difference between the standard errors and assume autocorrelation lengths of similar magnitude between the SOCATv5 gridded datasets located in the coastal and open ocean domains, respectively. Next we calculate the total error for each 30°x30° degree raster region r as:



$$\sigma_r = \sigma_{r,o} + \sigma_{r,c} \tag{2}$$

and scale, for each grid-cell in the overlap area, the weight given to the open ocean and coastal ocean local $pCO_2$ value by the standard error of each raster region:

$$pCO_{2,overlap} = (1 - \frac{\sigma_{r,o}}{\sigma_r}) \cdot pCO_{2,o} + (1 - \frac{\sigma_{r,c}}{\sigma_r}) \cdot pCO_{2,c} \tag{3}$$

To generate the final merged product we perform an additional smoothing using a 8x8 grid point running mean filter (roughly 200km by 200km at the equator).

## 3    Results and discussion

### 3.1    Large scale $pCO_2$ patterns along the coastal-open ocean continuum

The long term mean $pCO_2$ field at $0.25°$ resolution for $NN_{open}$ and $NN_{coast}$ is shown in Figure 4. In most oceanic regions,
the transition from open to coastal ocean occurs without steep gradients, particularly in the subtropics ($\sim 20°$N-$50°$N) of the northern hemisphere. However, exceptions exist in the tropics like the Peruvian upwelling system, the Namibian/Angolan coast in the South Atlantic and off Somalia and the Arabian Peninsula. Moreover, abrupt spatial gradients in $pCO_2$ have been observed in large river plumes such as that of the Amazon (Ibanhez et al., 2015) or on continental shelves influenced by large rivers. The identification of such gradients, however, results only from a first order visual inspection between the two products.
In what follows, we perform a quantitative analysis of the merging procedure and of the resulting $pCO_2$ fields in the overlap area.

Figure 5 reports the absolute $pCO_2$ difference in % between $NN_{coast}$ and $NN_{open}$ along the common overlap area relative to the mean partial pressure of the merged climatology. Figure 5 shows a clear latitudinal pattern with the lowest difference in the low and subtropical latitudes and the largest differences in the high latitudes, especially in the northern hemisphere. We find
in particular, that discrepancies are large in the newly added Arctic Ocean, but also in other seasonally ice-covered areas that have been previously described in $NN_{open}$ and $NN_{coast}$ publications (e.g. the Labrador Sea). One significant contributer to this difference might be that $NN_{coast}$ uses information about seaice in reconstructing the surface ocean pCO2. Acknowledging this discrepancy in seasonally ice-covered regions, we further focus our error analysis and products comparison on ice-free areas, based on the sea-ice product of Rayner et al. (2003). There are some exceptions to this general latitudinal trend consistent with
our first qualitative inspection, such as along the Pacific coastline of South America, the African coast in the South Atlantic and the Arabian Sea, i.e. the regions with steep gradients already identified above. Furthermore, a gradient of decreasing $pCO_2$ from the coast to the open ocean has been reported over the continental shelves of the Eastern US and Brazil (Laruelle et al., 2015; Arruda et al., 2015) and may exist in other regions as a consequence of the influence of rivers oversaturated in $CO_2$

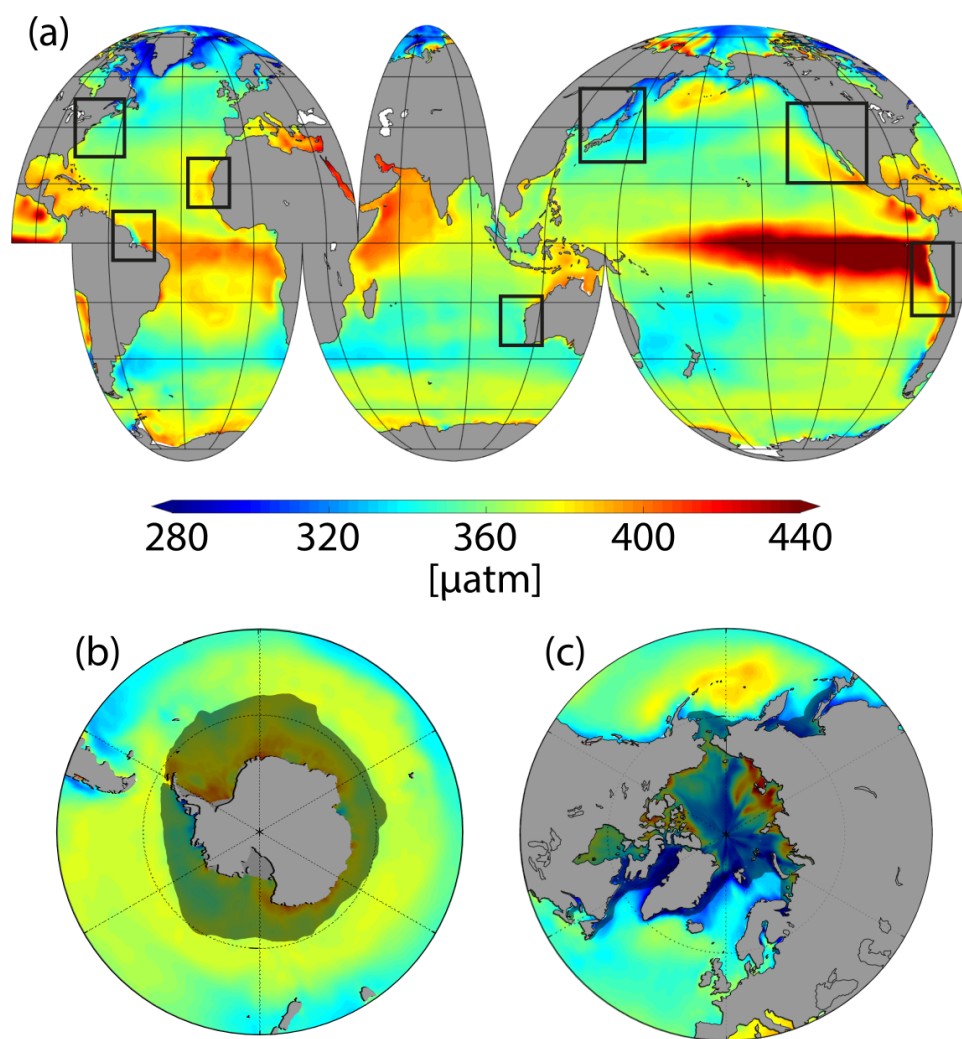

**Figure 4.** (a) climatological mean $p\mathrm{CO_2}$ of the merged product presented in this study. Panels (b) and (c) highlight the polar regions. Black Boxes in (a) illustrate regions that are further investigated in the regional analysis. Shaded areas in (b) and (c) delineate the maximum sea ice extend.

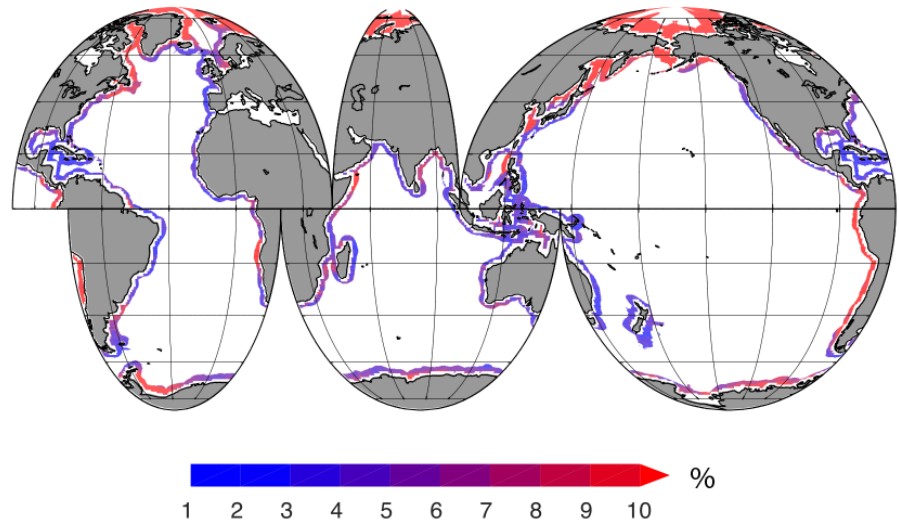

**Figure 5.** $pCO_2$ mismatch between $NN_{coast}$ and $NN_{open}$ in the overlap area relative to the mean $CO_2$ partial pressure of the merged product. Blue colors indicate a mismatch below 5%, whereas red colors indicate a mismatch of more than 5%.

combined with a limited estuarine filter (Laruelle et al., 2015). It is thus possible that the $pCO_2$ predicted by the coastal SOM-FFN are slightly skewed towards higher values in some regions because of presence of overall higher $pCO_2$ observations in the calibration data pool. While there is no clear basin-wide bias structure, systematic differences can be found regionally such as in the southeast Pacific Ocean and the Southern Ocean (south of 35°S). Overall, the largest relative differences are located in
5    the overlap areas of the Arctic Ocean.

In spite of clear regional discrepancies, the mean difference, that is to say the bias, between the two estimates in the overlap area remains close to 0 $\mu$atm when integrated globally (table 1), whether or not the comparison is limited to the locations where observations exist (table 1 columns 1-3). Furthermore, the mismatch between the two products is in the range of the mismatch between the individual products and the available observations in SOCATv5. This result is a consequence of the
10   neural network-based interpolation applied here at the global scale. In particular, the SOM-FFN is designed to minimize the mean squared error between available observations and the network output over the entire domain of application.

The global RMSE between $NN_{open}$ and $NN_{coast}$ as well as the SOCAT observations within the overlap area is in the range of previously reported global values by Landschützer et al. (2016) and Laruelle et al. (2017). In general, the spread between open ocean and continental coastal $pCO_2$ varies more than the spread between coastal estimates and SOCAT or between
15   open estimates and SOCAT, possibly indicating that the SOM-FFN method is having difficulties generalizing the $pCO_2$ in the coastal-open ocean continuum.



**Table 1.** Mean error analysis (bias and RMSE) within the overlap area between $NN_{coast}$ and $NN_{open}$ and the observations from the SOCATv5 dataset . The comparison is performed for the total overlap area, the area fraction where no observations exist and the area covered by observations. The bias and RMSE between the $pCO_2$ map products and the SOCATv5 open and coastal datasets are also reported.

|  | Coastal-open total | Coastal-open no obs. | Coastal-open colocated to obs. | Open-SOCAT | Coastal-SOCAT |
|---|---|---|---|---|---|
| Bias [$\mu$atm] | 0.6 | 0.6 | 0.6 | 0.7 | 1.5 |
| RMSE [$\mu$atm] | 36.4 | 36.9 | 20.0 | 18.3 | 26.8 |

## 3.2 Regional analyses of $pCO_2$ field

A more detailed analysis is performed in the overlap of several regions selected to encompass a wide variety of conditions. These regions, indicated in Figure 4, include three areas characterized by strong upwelling and offshore transport (Peruvian upwelling system, Canary upwelling system, US west coast) but contrasted data coverage, two data rich regions (Sea of Japan, US east coast), one region where seasonal data are scarce (West Coast of Australia), and a region characterized by strong river outflow (Amazon river plume).

In order to further investigate the role of existing observations in upwelling regions we first focus on the Canary upwelling system and the Peruvian upwelling system. These two regions are part of the Eastern Boundary Upwelling Systems and subject to many ecosystem stressors, such as ocean acidification or deoxygenation (Gruber, 2011). Therefore, monitoring the full aquatic continuum is essential in these regions. Both are characterized by strong upwelling and significant offshore transport of carbon rich water from depth (see e.g. Lovecchio et al., 2018; Franco et al., 2018) resulting in elevated $pCO_2$ levels exceeding atmospheric levels at the sea surface. Such values are consistent with observations in the Canary upwelling system (Figure 6) extracted from either the open ocean SOCAT dataset (Bakker et al. (2016), Figure 6b) or the coastal SOCAT dataset (Bakker et al. (2016)Figure 6c) and, consequently, the merged $pCO_2$ product (Figure 6a). Furthermore, the Canary upwelling system is well covered by both open ocean and coastal ocean observations. As a consequence - despite a few areas with larger differences - the overall mismatch between the coastal ocean and $NN_{open}$ (figure 6d) is in the range of their relative mismatch towards the observations (see figure 6e-f) and generally within $10\mu$atm.

In contrast to the Canary upwelling system, the Peruvian upwelling system shows a steep $pCO_2$ gradient between the offshore and near shore regions (Figure 7a), particularly just south of the equator. A closer inspection of the available observations (Figure 7b and c) reveals that, particularly in the near-shore domain at the equator, several of the few available observations of the sea surface $pCO_2$ indicate low partial pressures resulting in a low reconstructed coastal $pCO_2$, as already identified by Laruelle et al. (2017). The mismatch that results from the upscaling of the low $pCO_2$ data in the coastal domain is further reflected in the difference between the coastal and open ocean $pCO_2$ fields in the overlap area (figure 7 d). The mismatch between the open ocean and $NN_{coast}$ exceeds $30\mu$atm and is larger than the difference between the individual products and the observations (figures 7 e-f), suggesting that the disagreement between the open ocean and $NN_{coast}$ in the overlap area stems from their data treatment. The fewer existing coastal observations of low $pCO_2$ are extrapolated in space, spreading a potential

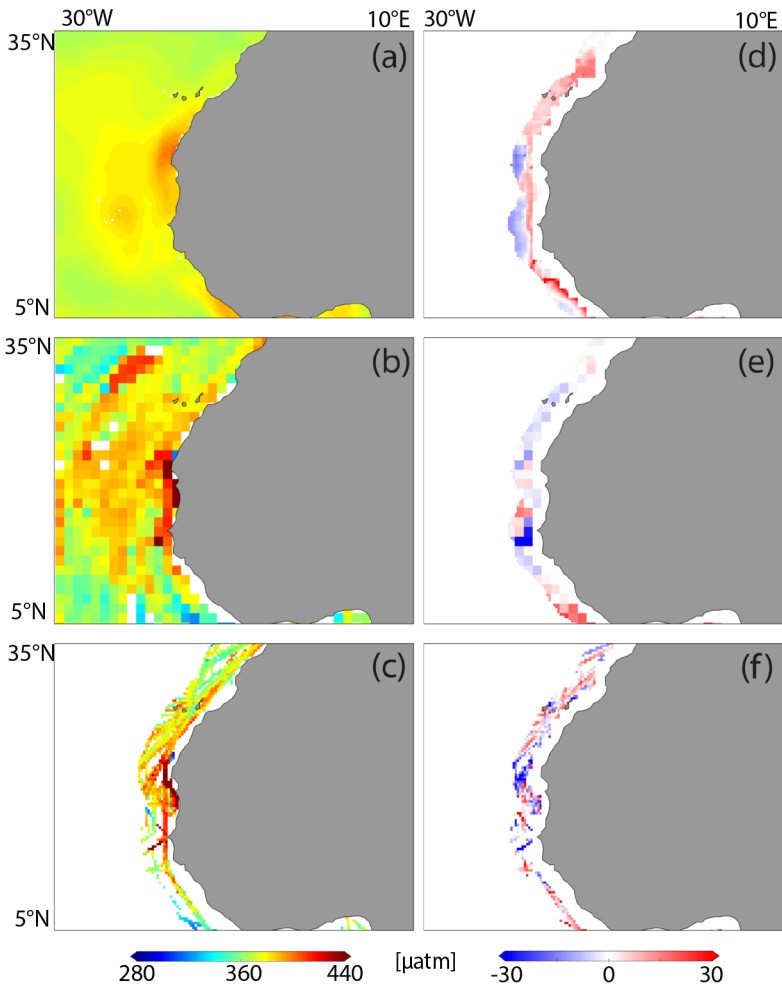

**Figure 6.** Mismatch analysis along the Canary upwelling region from 1998 through 2015 period. The climatological mean $pCO_2$ is reported for (a) the merged product, (b) all available SOCATv5 data for the open ocean, and (c) all coastal SOCATv5 data (as illustrated in Figure 1 for the global ocean). The $pCO_2$ mismatch is illustrated in (d) as the difference between $NN_{coast}$ and $NN_{open}$. Panel (e) reports the mismatch between the $NN_{open}$ and the SOCATv5 open ocean dataset along the overlap area while panel (f) reports the mismatch between the coastal product and the SOCATv5 coastal dataset along the overlap area.

Earth System Discussions
Science
Data

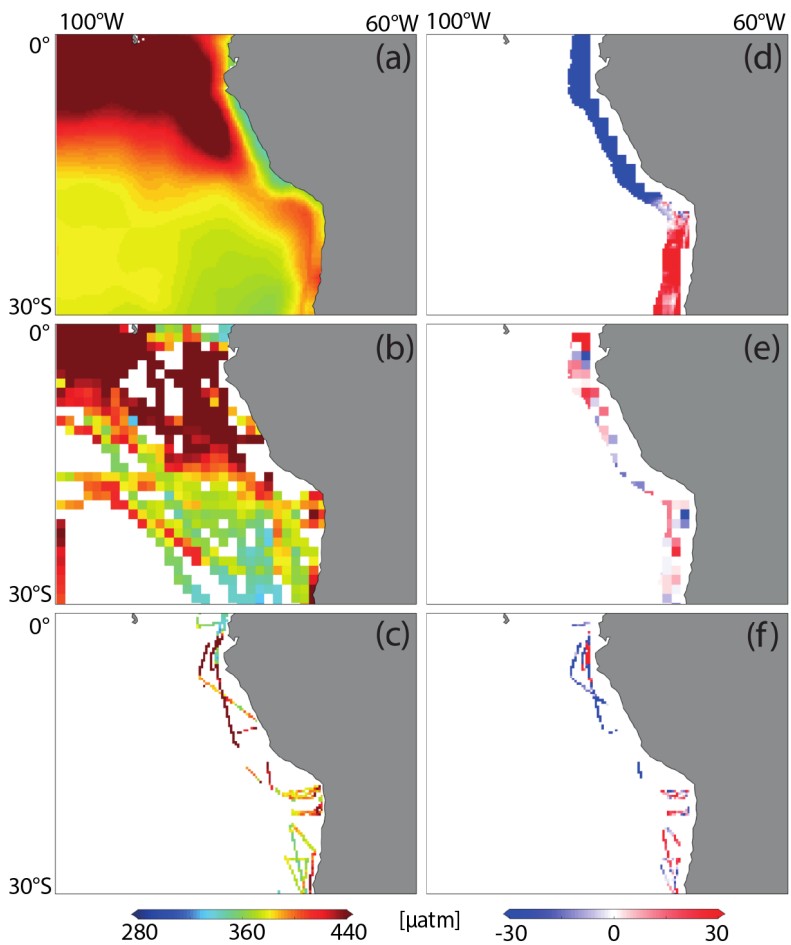

**Figure 7.** Like Figure 6 but for the Peruvian upwelling system

mismatch over a larger area. Likewise, the near-shore domain in the $NN_{open}$ is influenced by the high $CO_2$ partial pressures off-shore. This data sparsity and spatial heterogeneity is a further challenge for model evaluation Franco et al. (2018).

No steep $pCO_2$ gradient can be identified along the west coast of Australia in the merged product (Figure 8). The highest $CO_2$ partial pressures are found near shore along the Leeuwin current (Smith et al., 1991) and the lowest observed $pCO_2$ can be

5   found along the West Australian current. The area is spatially well covered both in the open and coastal ocean SOCAT datasets (Figure 8 b and c) and therefore the overall difference towards observed values remains among the smallest of all investigated regions. This is remarkable given the lack of seasonal observations, which will be discussed in the subsequent section. $NN_{open}$ and $NN_{coast}$ agree with each other spatially within 15 $\mu$atm (figure 8d), which is in the range of the mismatch between the individual products and the respective SOCAT observations (figures 8 e-f). Both products tend to overestimate the low $pCO_2$

10  towards the South of the domain. This is reflected in the positive mismatch towards the SOCAT observations (Figure 8 e and



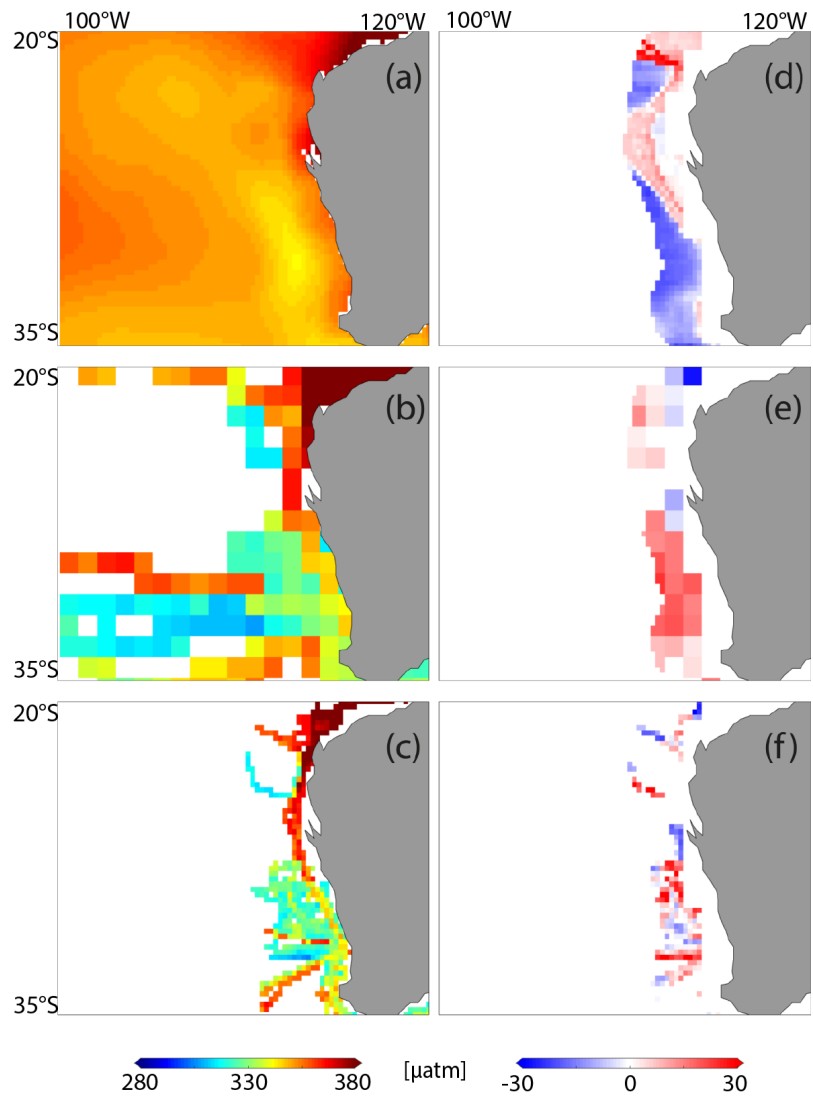

**Figure 8.** Like Figure 6 but for the Australian west coast

f) in the common overlap area where, the difference between the neural network estimates and the raw data exceeds 15 $\mu$atm for both products.

Observations in the Sea of Japan and adjacent Pacific Ocean suggest large variability in the $pCO_2$ with the lowest observed values just north of the Korean peninsula and the highest observed $pCO_2$ in the Yellow Sea (figures 9 b-c). Furthermore, low $pCO_2$ is also observed south of the island of Hokkaido. These large spatial variations in the $pCO_2$ are also visible in the merged $pCO_2$ product (figure 9a). A notable exception is the Korean Straight, where observations suggest a lower $pCO_2$ than reconstructed. The strong variability in the observed $pCO_2$ reflects the complex carbon dynamics in the Sea of Japan



(Chen et al., 1995; Park et al., 2006), which is also reflected in the larger mismatch between products and towards the SOCAT observations (figures 9 d-f). A better agreement between the neural network reconstructions and observations is found in the Pacific Ocean east of the Japanese islands, where the merged estimate also reveal a better agreement between $NN_{open}$ and $NN_{coast}$ (Figure 9 d) and low biases in the range of 5 $\mu$atm towards SOCAT observations (Figure 9 e and f).

Some of the best monitored regions spanning both coastal and near-shore open ocean can be found along the US coast (Fennel et al., 2008; Laruelle et al., 2015; Fennel et al., 2019). Indeed all 1x1° open ocean and almost all 0.25°x0.25° coastal pixels are filled with raw observations off the eastern US coastline. While the mean of all observed $pCO_2$ values from SOCAT (Figure 10b and c) suggests substantial regional variability, the merged estimate (Figure 10a) is, as a result of the neural network interpolation algorithm, substantially smoother. In particular, the lower latitudes (25-35°N, Figure 10e and f) are

well reconstructed by the neural network algorithms in both open and coastal ocean domains. Larger discrepancies however exist in the higher latitudes (35-45°N, Figure 10e and f). Landschützer et al. (2014) attributed this mismatch to the complex biogeochemical dynamics of the Gulf Stream region, where the $pCO_2$ is overestimated by both the open and coastal estimates. The smooth transition between coastal and open ocean in Figure 10a indeed suggests that the intensively surveyed US east coast aquatic continuum can be well reconstructed by combining the open ocean and coastal ocean $pCO_2$ datasets.

Similarly well monitored to the US east coast is the US west coast upwelling system, not the least because its variability is tightly linked to El Nino Southern Oscillation (see e.g. Lynn and Bograd, 2002; Frischknecht et al., 2015). Here, we find an overall good agreement between $NN_{coast}$ and $NN_{open}$. The agreement in the overlap area of the merged product (Figures 11d) is among the best reported globally. Interestingly, near shore, the merged estimate (Figure 11a) reveals a lower mean $pCO_2$ than suggested from both the open ocean and coastal ocean SOCAT datasets (figure 11 b and c). The small error compared

to the SOCAT observations suggests that this is not the result of the 2 products being in disagreement but might relate to the climatological nature of the merged product, which does not reflect the variable upwelling as a result of interannual variability linked to ENSO events.

    Finally, we investigate the spatial structure of the reconstructed $pCO_2$ from a region typically dominated by the freshwater outflow of a large river mouth, i.e. the Amazon outflow in the tropical Atlantic Ocean (Figure 12). Studies linking circulation

with the local $CO_2$ dynamics are sparse (Ibanhez et al., 2015; Lefevre et al., 2013). Very few observations exist, particularly in the near-shore region (Figure 12b-c). Nevertheless, studies suggest that the Amazon river outflow becomes a significant $CO_2$ sink when it mixes with ocean waters (Lefevre et al., 2010). The strong variance in observed $pCO_2$ (Bakker et al., 2016) provides a challenge for any algorithm to reconstruct the full $pCO_2$ field in such region. Nevertheless, both coastal and oceanic data products are in good agreement (Figure 12d) with the exception of the area under direct influence of Amazon River

outflow. This difference potentially stems from the $NN_{open}$ being unable to associate the $pCO_2$ variability observed in this area to the strong salinity gradients, which is better represented in the coastal ocean $pCO_2$ product. Both products show differences of similar magnitude when compared to the SOCAT observations (Figure 12e-f) and similar error structures as both products overestimate the $pCO_2$ in the northern and underestimate the $pCO_2$ in the southern sections of the overlap area.

    While global errors between the data products and observations remain low (see table 1), figures 6-12 show that, at the re-

gional scale, larger differences emerge. We therefore expend our standard error statistics as presented in table 2 for the selected



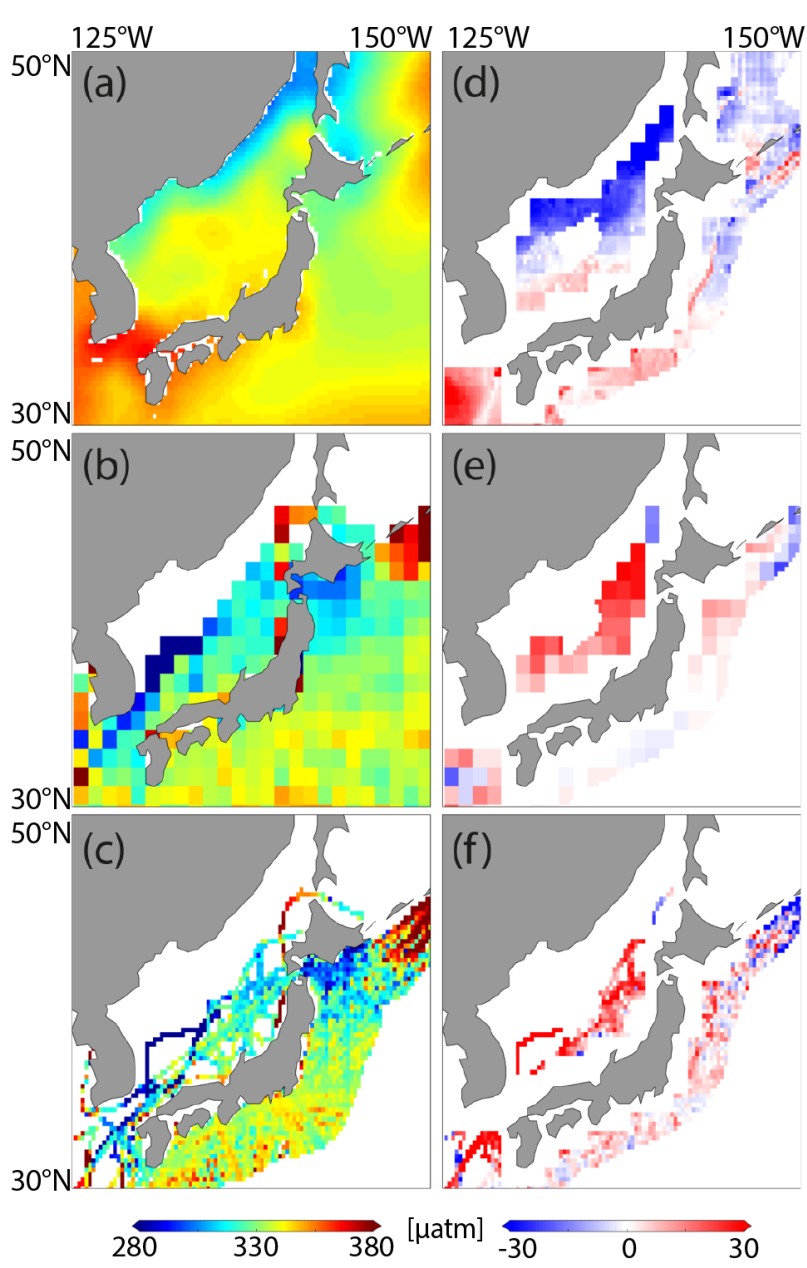

**Figure 9.** Like Figure 6 but for the Sea of Japan

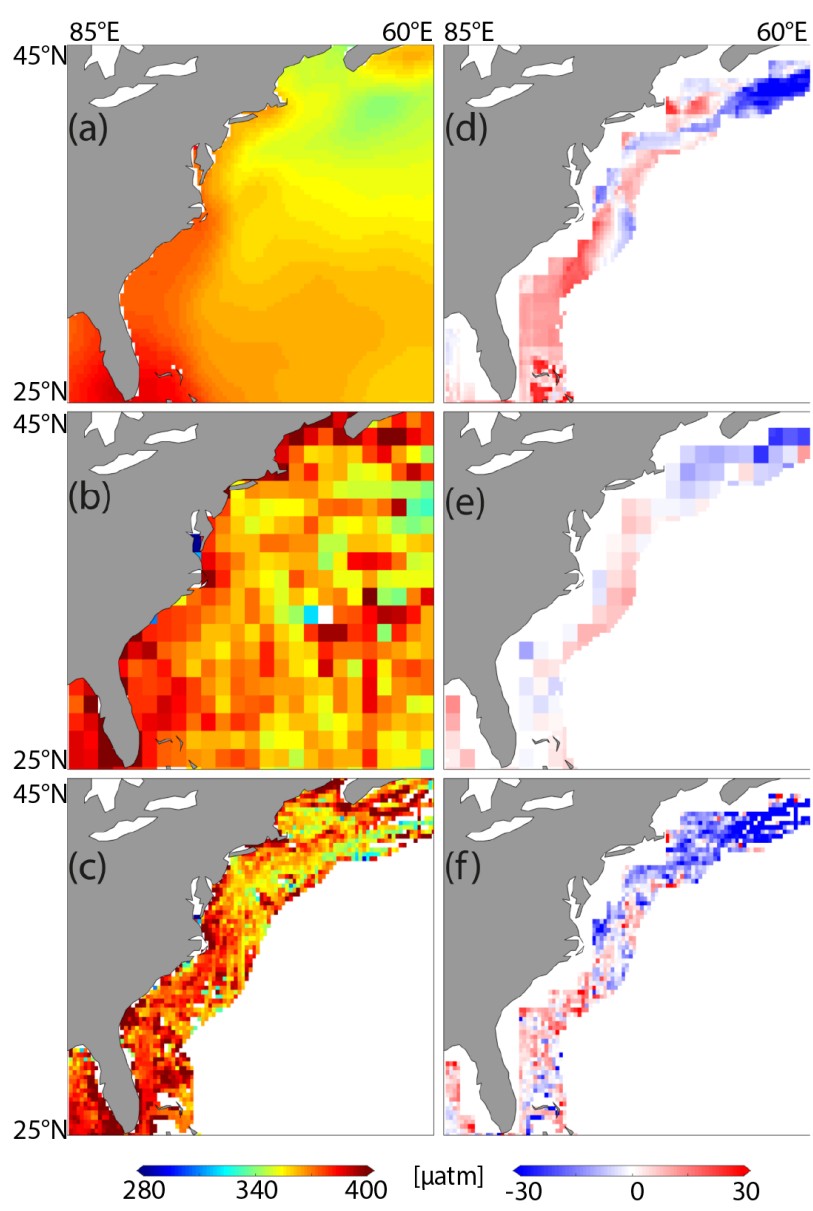

**Figure 10.** Like Figure 6 but for the US east coast



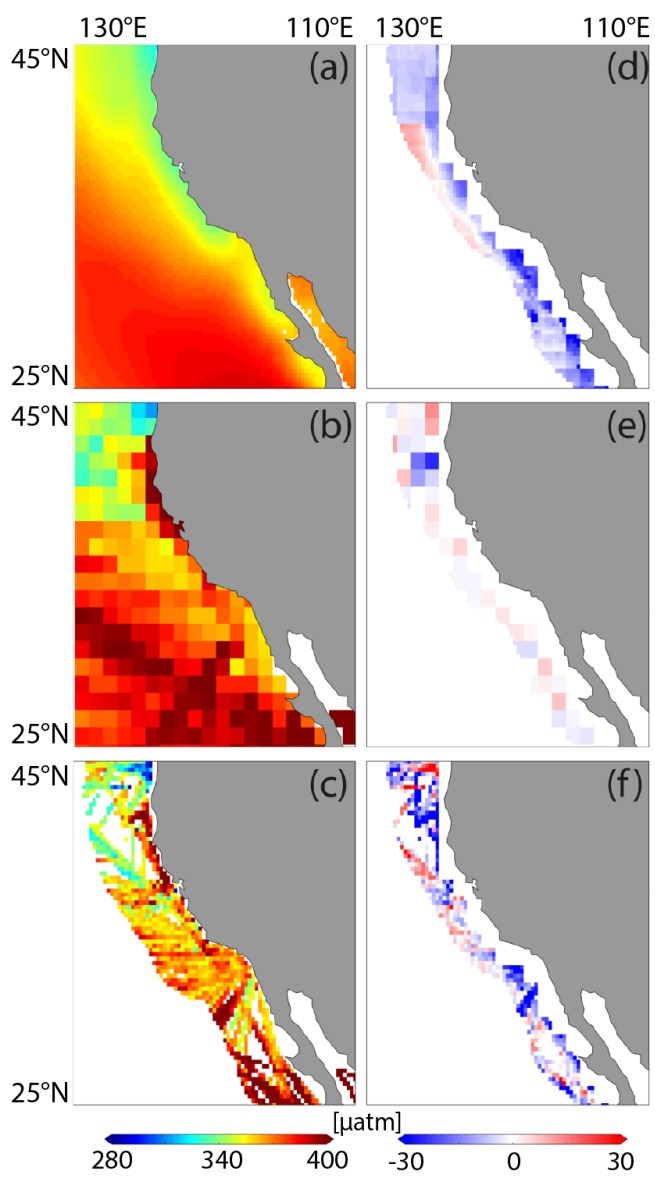

**Figure 11.** Like Figure 6 but for the US west coast

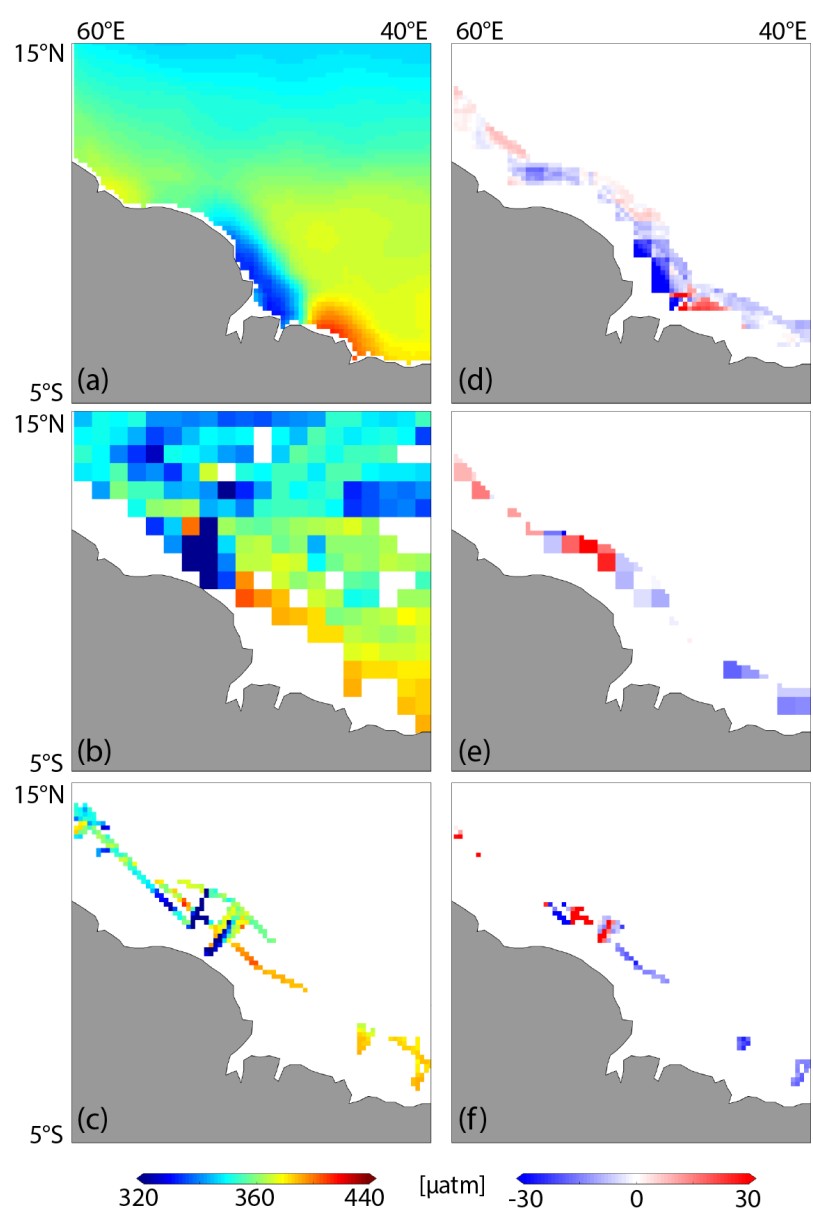

**Figure 12.** Like Figure 6 but for the Amazon outflow region in the tropical Atlantic Ocean



**Table 2.** Mean error analysis (bias and RMSE) within the overlap area between $NN_{open}$ and $NN_{coast}$ and the observations from the SOCATv5 dataset (Bakker et al., 2016) for 7 oceanic regions. The comparison is performed for the total overlap area, the area fraction where no observations exist and the area covered by observations. The biases and RMSE between $pCO_2$ products and SOCATv5 datasets are also reported for the open ocean and coastal ocean.

| Region | Coastal-open total bias (RMSE) [$\mu$atm] | Coastal-open no obs. bias (RMSE) [$\mu$atm] | Coastal-open colocated to obs. bias (RMSE) [$\mu$atm] | Open-SOCAT bias (RMSE) [$\mu$atm] | Coastal-SOCAT bias (RMSE) [$\mu$atm] |
|---|---|---|---|---|---|
| Canary upwelling system (5-35°N) | 3.6 (20.3) | 3.8 (20.5) | -1.0 (16.3) | -0.6 (16.3) | -1.3 (24.6) |
| Peru upwelling system (0-30°S) | -34.3 (80.6) | -34.3 (80.7) | -14.8 (42.0) | 2.2 (23.0) | -12.9 (49.0) |
| Australia west coast (20-35°S) | -3.4 (25.2) | -3.4 (25.3) | -7.6 (16.8) | 8.5 (17.4) | 4.1 (16.5) |
| Sea of Japan (30-50°N) | -3.5 (34.5 ) | -4.2 (35.8) | 2.4 (18.6) | 2.0 ( 16.5) | 4.5 ( 25.3) |
| US east coast (25-45°N) | 1.7 (26.0) | 2.4 (26.6) | -3.8 (21.1) | -0.1 (17.4) | -3.5 (27.9) |
| US west coast (25-45°N) | -7.5 (20.6) | -7.6 (20.7 | -6.5 (19.6) | 0.1 (13.7) | -7.0 (27.5) |
| Amazon outflow (5°S-15°N) | -5.5 (29.0) | -5.5 (29.0) | -0.5 (22.3) | 11.2 (37.9) | 14.8 (59.0) |

regions. Overall, we find at the regional level that the inter-product mismatch, represented by the bias, is substantially larger than in the global analysis but does not exceed ~8$\mu$atm with one prominent exception: the Peruvian upwelling system where the mismatch reaches 14.8 $\mu$atm. Here, the substantial disagreement between the two products results from the underestimation of the coastal observations in the overlap domain by the coastal ocean $pCO_2$ product already shown by Laruelle et al. (2017).

5    We find that the bias between $NN_{open}$ and $NN_{coast}$ in the overlap area are larger where they are not co-located to observations (Table 2). The error spread between $NN_{open}$ and $NN_{coast}$, represented by the RMSE, is likewise larger in areas where fewer observations exist (contrast column 1 and 2 in Table 2). Exceptions include the US east Coast and the West coast of Australia possibly linked to the larger mismatch of the individual products towards the respective SOCAT observations at these locations. Results from both products in the Amazon outflow region, in the US east coast for $NN_{coast}$ and in the west coast of Australia for

10   $NN_{open}$ show a larger bias towards the SOCAT observations than the respective inter-model bias, illustrating that both methods generalize well. This further suggests that the estimates are locally constrained by information outside the investigated domain, which is possible considering the spatial distributions of the biogeochemical provinces generated by the SOM.



### 3.3 Seasonality

A further analysis in the selected regions aims to investigate the seasonal differences in $pCO_2$ between the original data products, the merged product, and observations (Figure 13). In particular, we investigate the extent to which the mean biases reported above can be explained by seasonal differences in $pCO_2$ among the different products. To this end, we average all

months from 1998 through 2015 to create a seasonal climatology from our $pCO_2$ products, without correction to a nominal reference year. We repeat this procedure for the SOCAT datasets, likewise without any corrections but being aware that this could lead to a sampling bias in the observed climatology. This approach is justified because we lack knowledge about the short-term variability in the observed carbon cycle and it is thus unclear on how such a correction would improve the representation of the observed $pCO_2$ field.

In spite of the lack of seasonal sampling bias corrections, our analysis displays, for most regions, a close correspondence within a few $\mu$atm between open ocean and coastal ocean $pCO_2$ data from SOCAT within the overlap area (blues and yellow bars in Figure 13) with deviations mostly arising in the Peruvian upwelling system and the Amazon outflow regions where monthly differences can exceed 10 $\mu$atm. The good correspondence is expected to some degree because both datasets share a large fraction of the data. The analysis shows that the seasonality of the neural network-based on $NN_{open}$ and $NN_{coast}$ satis-

factorily reproduce the seasonal fluctuations obtained directly from the raw data, highlighting that the reconstructed seasonal cycle is well constrained by the existing observations. Monthly deviations between the products largely stay within 10 $\mu$atm. An exception is the Sea of Japan in boreal winter, where $NN_{open}$ overestimates the surface ocean $pCO_2$ values recorded in the SOCAT data. All but three of the selected regions have full seasonal data coverage. The three regions without full coverage are the West coast of Australia, the Amazon outflow region and the Peruvian upwelling system. Despite the lack of seasonal

observations along the West coast of Australia, both products agree well with regards to the seasonal cycle and differences stay within of 8-10$\mu$atm between the different products. Likewise, the otherwise good agreement between coastal ocean and open ocean estimate breaks down in the boreal summer in the Amazon outflow region, despite the lack of strong seasonality in the tropical latitudes.

The largest mismatch between data products and observations exist along the Peruvian upwelling system, where monthly

differences between open ocean and coastal ocean estimates exceed 40$\mu$atm. Both estimates however show similar seasonal variability. The seasonal analysis further reveals that from all investigated regions, the Peruvian upwelling system shows the largest monthly differences between open ocean and coastal ocean SOCAT observations, with e.g. mean differences in March exceeding 30$\mu$atm between the open ocean and coastal ocean SOCAT datasets (Bakker et al., 2016). Furthermore, the largest observed partial pressures in $NN_{open}$ appear in August where no data are available in the coastal ocean SOCAT dataset,

highlighting that $NN_{open}$ draws information from observations further away from shore during this month.

### 4   Data availability

The merged climatology (Landschützer et al. (2020), doi: 10.25921/qb25-f418) is available from NCEI OCADS and can be accesed via: https://www.nodc.noaa.gov/ocads/oceans/MPI-ULB-SOM_FFN_clim.html. $NN_{open}$ is available vie NCEI OCADS





**Figure 13.** Seasonal $pCO_2$ cycle for the seven regions discussed in the text and highlighted in the center map. The seasonal cycles include a comparison of the monthly mean SOCAT observations without any interpolation (blue and yellow bars) as well as the open ocean (blue line), coastal ocean (red line) and merged (magenta line) reconstructions based on the respective SOCAT observations.





and is accessible online https://www.nodc.noaa.gov/ocads/oceans/SPCO2_1982_present_ETH_SOM_FFN.html. $NN_{coast}$ description and dataset can be downloaded from the following url: https://www.biogeosciences.net/14/4545/2017/

## 5    Conclusions

In this analysis, we combined two recently published sea surface $pCO_2$ products, covering the open ocean and the coastal domain. While the spatial coverage of $NN_{open}$ includes all surface waters located further than $1°$ off the coast, the spatial coverage of the $NN_{coast}$ includes surface waters until 400km off the coast, leading to a roughly 300km wide overlap domain around the land surface. The common overlap area was used to compare both reconstructed $pCO_2$ estimates at regional to global scale and whether the observed agreement/disagreement is linked to data availability.

Our results show that, for most of the global ocean and particularly the subtropical latitudes in the northern hemisphere, $NN_{open}$ and $NN_{coast}$ agree well within the overlap domain. However, stronger differences exist in other parts of the world, particularly in the Peruvian upwelling system, the Arctic and Antarctic, the African coastline in the South Atlantic and the Arabian Sea, where fewer observations exist. In other regions without complete seasonal data coverage such as the west coast of Australia, however, both products compare well. We therefore conclude that the lack of data coverage in combination with biogeochemical complexity triggered by upwelling, river influx or seasonal ice coverage contribute both to the mismatch. Closer inspection reveals that for most of the overlap regions, the difference between the open ocean and coastal ocean estimates falls within the range of the difference between $NN_{open}$ and $NN_{coast}$ and the respective SOCAT dataset from which they were created. Therefore, the combined $pCO_2$ climatology is not only a step forward in including the full oceanic domain with all its complexity into carbon budget analyses, but also help identify areas where additional continuous observations are critically needed to close current knowledge gaps.

Another way forward to further reduce the bias between the coastal and open ocean estimates would be to reconsider the cut-off definition between the two domains. Data sparse and often strongly variable regions such as the Peruvian upwelling system are very sensitive to the data selected to generate the $pCO_2$ fields. The proposed overlap analysis here and particularly the RMSE analysis, further serves as a benchmark on how well we understand the coastal-to-open ocean continuum and its spatial variability and where we still lack essential measurements to close the gap between existing estimates. A next step should include the reduction of the mismatch between coastal and open ocean estimates in order to combine the two. This is an essential step towards an observation-driven global carbon budget. Closing such gap requires however close collaborations between open ocean and coastal ocean carbon cycle scientists in the future and be considered of high importance.

Finally, we introduced a new concept where we can locally evaluate the upscaling of existing measurements based on a common overlap region. In this study, we focused on mean differences and seasonal climatologies at regional and global scales. We find an encouraging agreement between seasonal cycles which gives us confidence that the existing products might be suitable to be applied to study lower frequency signals such as trends and interannual variability. Understanding of how differences in trends and inter-annual variabilities between the coastal and open oceans emerge and how they are linked to data availability should be a next step. Such analysis is essential to gain confidence in observational constraints and to find ways to



further improve them in order to close the global carbon budget based on observations and provide data products form model benchmarking. Our approach can also be used to compare other overlapping datasets at a time when advanced interpolation techniques are yielding more and more oceanic data products with different spatial extensions and boundaries. Our study is therefore an important step towards a truly representative global ocean observation-based $CO_2$ product that includes all ocean

5  domains.

*Competing interests.* We declare no competing interests

*Acknowledgements.* PL is supported by the Max Planck Society for the advancement of Science. The research leading to these results has received funding from the European Community's Horizon 2020 project under grant agreement no. 821003 (4C). GGL is research associate of the F.R.S-FNRS at the Universite Libre de Bruxelles. PR received funding from the VERIFY project from the European Union Horizon

10  2020 research and innovation program under grant agreement No. 776810. This study benefited from discussions with K. Six from the Max Planck Institute for Meteorology. The Surface Ocean CO2 Atlas (SOCAT) is an international effort, supported by the International Ocean Carbon Coordination Project (IOCCP), the Surface Ocean Lower Atmosphere Study (SOLAS), and the Integrated Marine Biogeochemistry and Ecosystem Research program (IMBER), to deliver a uniformly quality-controlled surface ocean CO2 database. The many researchers and funding agencies responsible for the collection of data and quality control are thanked for their contributions to SOCAT.



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
