# Peer review of "A uniform pCO2 climatology combining open and coastal oceans"

_Earth System Science Data, 2020_

## Referee Comment (RC1) · Rik Wanninkhof (Referee) · 22 May 2020

Reviewer Rik Wanninkhof, NOAA/AONL A uniform pCO2 climatology combining open and coastal oceans Peter Landschützer, Goulven G. Laruelle" Alizee Roobaert, and Pierre Regnier

The is a nice descriptive paper providing the procedures of merging the coastal pCO2 NN data from Laruelle et al. 2017 with the global fields of Landschützer et al. 2016. It gives an overview of the means of merging, and then provides an extensive analysis of the differences in the region of overlap using several coastal locations as examples. Writing style, syntax and grammar are very good and procedures are clearly described. Figures are of good quality but I wished there would be a way the more clearly show

the coastal area that shows up as a thin multi-colored rind in the figures.

The paper is an important contribution in documenting the procedures and outcomes of the combining exercise, and shows, on the whole, a consistent final product. Laruelle et al. 2017 mentioned that the products could be "readily merged". As this paper aptly describes the merging is not "readily done" but requires specific procedures, assumption and approaches which are well detailed in this manuscript. My comments below should not be considered a requirement for changing the manuscript, that seems good as is, but rather issues that came to mind while reading the paper. It therefor does not require a point by point rebuttal.

General comments - There should be some indication of how many observations there really are in the coastal region (and Open ocean overlap). % of pixels with observations (where the pixel is the 0.25 degree monthly "grid box" for the time period) is a good metric for each of the 30ËŽ regions investigated. - Different predictors are used for the coastal product and the open ocean dataset. E.g. Coastal uses wind and bathymetry (and sea ice); while the open ocean uses mixed layer depth (MLD). Is there any estimate how different the nn outputs are? That is, perhaps some mention if the different predictors influence the comparison between open ocean and coastal. In particular, what is the effect of not using MLD in the coastal product when we know large parts of the broad Western shelves are strongly stratified for part of the year? - What is not empathized is that in the overlap region the pCO2 observations used in coastal and open ocean products are exactly the same (I believe). - Is the data quality for the coastal data lower than for the open ocean? And, if so, does this have an effect (That is, I believe that that are more SOCAT "C" cruises in the coastal than in the open ocean).

Specific comments Page 1. Line 9 "This also illustrates the potential of such analysis to inform the measurement community about the locations where additional measurements are essential to better represent the aquatic continuum": This is also mentioned in the conclusions but I do not see clear evidence of how this is the case. Page 3. Line

5 "whereas Roobaert et al. (2019) suggests that this difference stems from the uneven latitudinal distribution of surface areas between coastal and open ocean but that adjacent open and coastal regions behave similarly.": I don't understand this. Page 3, line 15. "As a signiïficant fraction of this CO2 outgassing derived from terrestrial carbon inputs likely takes place near the coast or across the coastal-open ocean transition,": I believe that the working assumption is that this outgassing occurs in the southern hemisphere far away from the rivers (due to slow oxidation of riverine supplied terrestrial organic matter).

Page 4: It would be illustrative to show a map of the different provinces for coastal and open ocean (I know the boundary are not fixed but they do not vary that much ) Page 4 line 20 "Firstly, we replaced the mixed layer depth proxy of the NNopen from de Boyer Montegut et al. (2004) to the Argo based MIMOC product": a. How much difference does this make?; and b. If it is purely ARGO based it will be for water depths > 1200 m so much of the open ocean coastal overlap would not have good MLD.

Page 7. Line 14 "N is the number of available gridded data from SOCATv5 available in a given 30x30 raster box and the subscript I refers to either NNopen or NNcoast": This information would be of interest as a table for each 30 by 30 region Page 8. Line 17 "Figure 5 reports the absolute pCO2 difference in % between NNcoast and NNopen along the common overlap area relative to the mean partial pressure of the merged climatology.": Including this in a table for each province or 30 by 30 region along with the st deviation would be illustrative. Table 1- providing the % of coastal-no obs. And % coastal-open collocated would be of interest.

Fig 6. Providing the standard deviation of the mismatch shown d,e,f as extra panels would be of interest. Fig 7- 12 repeating the legend rather than stating "like Fig 6" will make reading the paper a bit easier Page 13. Line 5 "The area is spatially well covered both in the open and coastal ocean SOCAT datasets": It would be worthwhile to quantify what "well covered means" . Page 15. "Some of the best monitored regions spanning both coastal and near-shore open ocean can be found along the US coast

(Fennel et al., 2008; Laruelle et al., 2015; Fennel et al., 2019)": Perhaps include reference to "Signorini, S. R., Mannino, A., Najjar, R. G., M., F. M. A., Cai, W.-J., Salisbury, J., Wang, Z. A., Thomas, H., and Shadwick, E. H.: Surface ocean pCO2 seasonality and sea-air CO2 flux estimates for the North American east coast, J. Geophys. Res., 118, doi:10.1002/jgrc.20369, 2013."

Page 15: "climatological nature of the merged product, which does not reflect the variable upwelling as a result of interannual variability linked to ENSO events.": Could this be verified by looking at the standard deviation? Page 17. Figure 10 The N-S spatial trend in panels d-f is pretty apparent. While it is alluded to in the text the description seems a bit vague.

───────────────────────────────

---

## Referee Comment (RC2) · Anonymous Referee #2 · 4 Jul 2020

The reviewer enjoyed this article very much because the authors described how they merged open and coastal ocean pCO2 mapped climatology. The reviewer also observed that writing nature is very clear and good and procedures that they did are very clearly described. It is however the reviewer would like to suggest some to improve this article, therefore this article can be published ESSD after minor revision as stated below.

1, Page 4 line 4- On the data treatment about the overlapping area: The authors defined the open region and the coastal region as "covering broadly the open ocean at a distance of 1ÂŮę off the coast and, the second dataset, by Laruelle et al. (2017), covering the coastal domain plus the adjacent open ocean up until 400km away from the shoreline". And in page 6 line 2 the authors stated "landward limit of the NNopen

[Figure]

is located on average at around 1◦ (or roughly 100km) offshore". As the authors know 1 degree latitude is almost 110.6 km to 111.7 km but 1 degree longitude depends on latitude and varied from 111.2 km to zero. Therefore the authors should make clear how they define and treat the data as the open ocean. 2, page 7. Figure 3 is important to understand how the authors merged the open ocean product and the coastal region product. Therefore it might better to enlarge this figure 3. The reviewer also suggests adding a numerical table to show an example of how they merged. 3, page 10 In the Figure 5, the maximum of a color bar of mismatch percent means that clear red indicates exceed 10 %. The reviewer suggests extending this color bar at least 15 % or 20 % to clearly show the regions where the mismatch is large because a smaller mismatch region does not need to highlight but a larger mismatch region should be highlighted. 4, Page 14 line 3. The authors discussed about Sea of Japan. It is however this region is a marginal sea and it not appropriate to compare NNopen and NNcoast here because the Sea of Japan might be included into coastal region following 400 km definition from the Japanese coast and Korean/Russian coast. Furthermore, there are probably no observed data at the Korean/Russian side based on Figure 9 ( c ). Therefore it is better to delete this part from this article. 5, Figure 6,7,8,9,10,11,12: In (d)(e)(f) of these 7 figures, it is a little bit difficult to see the differences. Especially to distinguish difference zero region and no data region because the authors assigned no fill to both regions. Please re-draw these figures. 6, P21 line 19- The authors stated that "Despite the lack of seasonal observations along the West coast of Australia, both products agree well with regards to the seasonal cycle and differences stay within of 8-10$\mu$atm between the different products.". The reviewer observed in figure 13 that in these three regions NNopen and NNcosat products showed a minimum or a maximum although there are no observed data at the time of a minimum or a maximumãĂĄeg. a minimum in September on the west coast of Australia. The reviewer cannot understand how NNopen and NNcosat products there were produced and showed a minimum/maximum. Please explain this. 7, Page 21 line 17- The authors stated that "Therefore, the combined pCO2 climatology is not only a step forward in including the

full oceanic domain with all its complexity into carbon budget analyses, but also help identify areas where additional continuous observations are critically needed to close current knowledge gaps.". The reviewer completely agree this statement and would like to suggest to add some recommendations explicitly from the authors to the community about areas where additional continuous observations are critically needed to close current knowledge gaps. If the authors do so, the contribution of this article to the community will increase much. End of comments.

---

## Author Comment (AC1) · 30 Jul 2020

We would like to thank reviewer#1 Rik Wanninkhof for the thoughtful comments and suggestions. In the following we will respond (in italics) to each reviewer comment (printed in bold font) individually

**Reviewer Rik Wanninkhof, NOAA/AONL A uniform pCO2 climatology combining open and coastal oceans Peter Landschützer, Goulven G. Laruelle,, Alizee Roobaert, and Pierre Regnier**

**R#1: The is a nice descriptive paper providing the procedures of merging the coastal pCO2 NN data from Laruelle et al. 2017 with the global fields of Landschützer et al. 2016. It gives an overview of the means of merging, and then provides an extensive analysis of the differences in the region of overlap using several coastal locations as examples. Writing style, syntax and grammar are very good and procedures are clearly described. Figures are of good quality but I wished there would be a way the more clearly show the coastal area that shows up as a thin multi-colored rind in the figures. The paper is an important contribution in documenting the procedures and outcomes of the combining exercise, and shows, on the whole, a consistent final product. Laruelle et al. 2017 mentioned that the products could be "readily merged". As this paper aptly describes the merging is not "readily done" but requires specific procedures, assumption and approaches which are well detailed in this manuscript. My comments below should not be considered a requirement for changing the manuscript, that seems good as is, but rather issues that came to mind while reading the paper. It therefor does not require a point by point rebuttal.**

*Response: Many thanks for the overall positive assessment of our study and the helpful comments we received. While the reviewer does not ask for a detailed rebuttal, we took this opportunity to provide a point-by-point response describing how we have taken the referee suggestions into account, because we are eager to improve our manuscript and found many of the reviewer's suggestion very useful. We agree with the reviewer that the coastal and overlap bands are somewhat hard to see in Figures 1, 2 and 5 given the global projection we chose (due to the global nature of our study). We have thus tried alternative ways to display the coastal and overlap regions and found that the equidistant projection without longitude/latitude mesh lines offers the best visualization of all coastal features. We have illustrated this below where (a) represents the original version and (b) the new equidistant projection. We therefore adjusted Figures 1, 2 and 5 accordingly.*

[Figure]

*(a)*

*(b)*

**R#1: General comments - There should be some indication of how many observations there really are in the coastal region (and Open ocean overlap). % of pixels with observations (where the pixel is the 0.25 degree monthly "grid box" for the time period) is a good metric for each of the 30 regions investigated.**

*Response: As both the coastal and open ocean product rely on the gridded SOCAT data, we have now provided this information in the respective methods section, however, unlike suggested by the referee, we have (also as indicated below in response to other comments) refrained from providing a table with the error statistics (bias and standard deviation) and the number of observations for all 30x30 regions since this equates to 72 regions of which 57 are occupied. We believe that this would be a very large and cryptic table with little use to most readers. Hence we thought of alternative ways to display this information and opted for a box-whisker plot which, in our opinion best shows the proposed metrics. We therefore introduce the following new plot (new figure 4 in*

*the revised manuscript) in our revised manuscript instead of a table that would summarizes the number of data, std and mean difference of coastal ocean and open ocean product for the 30x30 regions*

*Additionally, we added the following text to the methods section: "Substantial differences exist between the mean difference and standard deviations of NNopen and NNcoast and the respective measurements from the SOCAT database within each 30x30 degree raster. Figure 4 illustrates these differences. While both NNopen and NNcoast have a near 0 bias for the mean differences, some rasters show differences exceeding 15µatm. While more variability appears in NNcoast, this can largely be explained by to the overall smaller number of gridded measurements. The larger number of gridded measurements in NNopen is a result from the division of the 1x1 degree cells into 16 quarter degree boxes. Therefore, we reduce the number of effective degrees of freedom for the open ocean by 16."*

[Figure]

*Caption: Box-Whisker plot of the mean difference (top), standard deviation (middle) and number of 0.25° pixels occupied with measurements (bottom) in the common overlap area for each 30°x30° box used for merging $NN_{open}$ and $NN_{coast}$.*

**R#1: Different predictors are used for the coastal product and the open ocean dataset. E.g. Coastal uses wind and bathymetry (and sea ice); while the open ocean uses mixed layer depth (MLD). Is there any estimate how different the nn outputs are? That is,**

**perhaps some mention if the different predictors influence the comparison between open ocean and coastal. In particular, what is the effect of not using MLD in the coastal product when we know large parts of the broad Western shelves are strongly stratified for part of the year?**

*Response: Besides this study, there is no quantitative assessment of the difference between both products. The reviewer is correct in that the products are different in the use of predictor data. We believe this remark best fits in the conclusions section of the manuscript, hence we have added a paragraph discussion these differences. This paragraph reads:*

*"Additionally, methodological differences between NNopen and NNcoast, such as differences in predictor data result in local differences, e.g. in ice covered regions where NNcoast relies on sea-ice as predictor or shallow, stratified waters, where mixed layer depth serves as important proxy in NNopen"*

**R#1: What is not empathized is that in the overlap region the pCO2 observations used in coastal and open ocean products are exactly the same (I believe). - Is the data quality for the coastal data lower than for the open ocean? And, if so, does this have an effect (That is, I believe that that are more SOCAT "C" cruises in the coastal than in the open ocean).**

*Response: The data in the overlap area are fairly identical, however there is a difference in the resolution of the gridded SOCAT data (which is illustrated in Figures 6 onwards panels b, c, e and f, as well as in the new figure introduced above). Indeed, the resolution of NNopen is 1 degree while the resolution of NNcoast is ¼ degree, which certainly influences the reconstruction. The difference in data quality is an interesting aspect, however we believe such an investigation is beyond the scope of this study, as it would require to check the individual cruises and how they feed into the gridded SOCAT gridded products. Furthermore, we believe that the uncertainty from extrapolating the observations over several hundreds of kilometers in distance contributes more to the overall uncertainty (compared to the 2µatm uncertainty from flag A and B data compared to 5µatm uncertainty from flag C data). Nevertheless, we have mentioned in the text that the gridded SOCAT data comprise of observations that received a flag A-D and therefore a potential uncertainty of 2-5µatm results from the measurement uncertainty.*

*In particular we added in the methods section: "The gridded SOCAT data consist of measurements that received a quality flag of D and lower, illustrating a measurement uncertainty within 5 µatm."*

**R#1: Specific comments Page 1. Line 9 "This also illustrates the potential of such analysis to inform the measurement community about the locations where additional measurements are essential to better represent the aquatic continuum": This is also mentioned in the conclusions but I do not see clear evidence of how this is the case.**

*We have rephrased this statement on page 1 to: "This also illustrates the potential of such analysis to highlight where we lack a good representation of the aquatic continuum and future research should be dedicated."*

*Regarding the sentence in the conclusion section, we expanded upon this statement to provide explicit recommendations based on the findings of this manuscript. In particular, we mentioned the Peru upwelling system and the high latitude regions, since we face a critical monthly difference between open ocean and coastal ocean reconstructions (see Figures 5 and 13), and we believe that this huge gap cannot be closed by improving the methods, but only by observing the field pCO2.*

*We therefore added: "The overlap analysis proposed here and particularly the Percent mismatch and RMSE analysis, further serves as a benchmark on how well we understand the coastal-to-open*

*ocean continuum and its spatial variability and where we still lack essential measurements to close the gap between existing estimates, such as e.g. the Peruvian upweling system or the seasonally ice-covered high latitude regions, in particular the Arctic Ocean"*

**R#:1Page 3. Line 5 "whereas Roobaert et al. (2019) suggests that this difference stems from the uneven latitudinal distribution of surface areas between coastal and open ocean but that adjacent open and coastal regions behave similarly.": I don't understand this.**

*Response: We rephrased this to: "... whereas Roobaert et al. (2019) suggests that adjacent open and coastal regions behave similarly."*

**R#1: Page 3, line 15. "As a significant fraction of this CO2 outgassing derived from terrestrial carbon inputs likely takes place near the coast or across the coastal-open ocean transition,": I believe that the working assumption is that this outgassing occurs in the southern hemisphere far away from the rivers (due to slow oxidation of riverine supplied terres- trial organic matter).**

*Response: Only a small amount of the riverine derived CO2 outgases in the Southern Ocean (compared to the large outgassing of natural carbon resulting from the upwelling of old carbon rich waters - see e.g. Figure 1b in Gruber et al 2009, "Oceanic sources, sinks, and transport of atmospheric CO2", Global Biogeochemical Cycles). The largest river outgassing fluxes – according to the work of Gruber et al 2009 and Mikaloff-Fletcher et al 2007 (Inverse estimates of the oceanic sources and sinks of natural CO2 and the implied oceanic carbon transport, Global Biogeochemical Cycles, 21, GB1010.) take place in the Northern hemisphere where most river input are delivered into the coastal ocean. This statement further refers to the work of Regnier et al 2013 (Figure 1a in "Anthropogenic perturbation of the carbon fluxes from land to ocean", Nature Geosciences) who do show that the Land Ocean Aquatic Continuum plays a significant role in redistributing carbon from riverine input. No changes have been made in the manuscript.*

**R#1: Page 4: It would be illustrative to show a map of the different provinces for coastal and open ocean (I know the boundary are not fixed but they do not vary that much)**

*Response: Many thanks for this suggestion. These province maps however are already introduced in Landschützer et al 2014 and Laruelle et al 2017. We have now mentioned in the text that these province maps can be found in these respective manuscripts.*

*In particular we added to the second paragraph in the methods section: "These provinces are illustrated in Landschützer et al 2014 and Laruelle et al 2017"*

**R#1: Page 4 line 20 "Firstly, we replaced the mixed layer depth proxy of the NNopen from de Boyer Montegut et al. (2004) to the Argo based MIMOC product": a. How much difference does this make?; and b. If it is purely ARGO based it will be for water depths > 1200 m so much of the open ocean coastal overlap would not have good MLD.**

*Response: a) we noted in the text, lines 21-22: "while the error statistics of the method remain nearly unchanged". We understand however, that this is fairly vague, hence we expanded a little further and wrote: "We tested the impact of this change and found that SOCAT observations are reconstructed bias free with a root mean squared error of less than 20µatm similar to Landschützer et al 2016"*

*b) This was a mistake on our end: The MIMOC MLD product is not entirely ARGO based, but combines (quoting from Schmidtko et al 2012): "All available quality- controlled profiles of temperature (T) and salinity (S) versus pressure (P) collected by conductivity-temperature-depth*

*(CTD) instruments from the Argo Program, Ice-Tethered Profilers, and archived in the World Ocean Database are used". We have corrected this in the text. For a detailed view of the profiles used and a comparison to other products such as de Boyer Montegut et al 2004, we can refer the referee to the original publication about the MIMOC mixed layer depth product: Schmidtko, S., G. C. Johnson and J. M. Lyman, 2013. MIMOC: A Global Monthly Isopycnal Upper-Ocean Climatology with Mixed Layers. Journal of Geophysical Research, 118, in press, doi: 10.1002/jgrc.20122.*

*We have now removed "Argo based" from the text.*

**R#1: Page 7. Line 14 "N is the number of available gridded data from SOCATv5 available in a given 30x30 raster box and the subscript I refers to either NNopen or NNcoast": This information would be of interest as a table for each 30 by 30 region**

*Response: We understand the interest in such a table, however, given that there are 12x6 such raster boxes (although not all are covered by both products), this table would be huge and would provide little information compared to its dimension. We therefore decided to introduce the error metric figure above to inform the reader (see comment 2 above). The number of 0.25° measurements is displayed in the lower panel.*

**R#1: Page 8. Line 17 "Figure 5 reports the absolute pCO2 difference in % between NNcoast and NNopen along the common overlap area relative to the mean partial pressure of the merged climatology.": Including this in a table for each province or 30 by 30 region along with the st deviation would be illustrative. Table 1- providing the % of coastal-no obs. And % coastal-open collocated would be of interest.**

*Response: We have included these error metrics for each 30x30 region in a new figure (see comment 2 above). We have, however decided to use absolute differences towards the actual measurements and std instead of % error in this case as open ocean and coast may be better comparable this way and since these are the metrics used for the merging. We further believe that figure 5 (now figure 6 in the revised manuscript) clearly illustrates the mismatch in % more refined in space (i.e. for each 0.25° grid box)*

**R#1: Fig 6. Providing the standard deviation of the mismatch shown d,e,f as extra panels would be of interest.**

*Response: Displaying the standard deviation of the mismatch in time as additional panel in the map is problematic for 2 reasons. Firstly, we believe that the spatially refined std is not always very meaningful for all chosen regions (with the exception of data rich regions, e.g. of the US coast, where repeat occupations exist) since very few ¼ degree pixels are occupied more than once in time. This is illustrated by the Amazon river outflow region below (color axis in µatm). Secondly, our figures already consist of 6 panels and we are afraid to "overload" the manuscript with figures that way. Instead we provide the RMSE in table 2 of the original manuscript for each region as we believe this provides an equally meaningful metric for the entire region.*

[Figure]

*Caption: Standard deviation of the mismatch as illustrated for the coastal ocean observations within the Amazon outflow region.*

**R#1: Fig 7- 12 repeating the legend rather than stating "like Fig 6" will make reading the paper a bit easier**

*Response: we have now repeated the legend for all figures.*

**R#1: Page 13. Line 5 "The area is spatially well covered both in the open and coastal ocean SOCAT datasets": It would be worthwhile to quantify what "well covered means" .**

*Response: We agree that the term "well covered" was not clear. In this particular case we rephrased to "… spatially covered both in the open and coastal …"*

**R#1: Page 15. "Some of the best monitored regions spanning both coastal and near-shore open ocean can be found along the US coast (Fennel et al., 2008; Laruelle et al., 2015; Fennel et al., 2019)": Perhaps include refer- ence to "Signorini, S. R., Mannino, A., Najjar, R. G., M., F. M. A., Cai, W.-J., Salisbury, J., Wang, Z. A., Thomas, H., and Shadwick, E. H.: Surface ocean pCO2 seasonality and sea-air CO2 flux estimates for the North American east coast, J. Geophys. Res., 118, doi:10.1002/jgrc.20369, 2013."**

*´Response: We have now added the additional reference in the revised manuscript*

**R#1: Page 15: "climatological nature of the merged product, which does not reflect the variable upwelling as a result of interannual variability linked to ENSO events.": Could this be verified by looking at the standard deviation?**

*Response: We believe that this would require more research than looking at the Standard deviation and is beyond the scope of this study. Nevertheless, we also note that the formulation was not entirely clear. Hence we rephrased our sentence into: "The small error compared to the SOCAT observations suggests that this is not the result of the 2 products being in disagreement but might relate to changes in upwelling as a result of interannual variability linked to ENSO events that are not well captured by the merged product."*

**R#1: Page 17. Figure 10 The N- S spatial trend in panels d-f is pretty apparent. While it is alluded to in the text the description seems a bit vague.**

*Response: We have now added extra emphasis to this difference*

*We added "Landschützer et al. (2014) attributed a larger mismatch to the complex biogeochemical dynamics of the Gulf Stream region, where the measured pCO2 is underestimated by both the open and coastal products. The strong mesoscale dynamics and the influence of the cold Labrador current in this region are not well represented in the rather coarse 0.25° NNcoast and 1° NNopen products"*

---

## Author Comment (AC2) · 30 Jul 2020

We would like to thank reviewer#2 for the thoughtful comments and suggestions. In the following we will respond (in italics) to each reviewer comment (printed in bold font) individually

**R#2: The reviewer enjoyed this article very much because the authors described how they merged open and coastal ocean pCO2 mapped climatology. The reviewer also observed that writing nature is very clear and good and procedures that they did are very clearly described. It is however the reviewer would like to suggest some to improve this article, therefore this article can be published ESSD after minor revision as stated below.**

*Response: Many thanks for the positive evaluation of our manuscript*

**R#2: 1, Page 4 line 4- On the data treatment about the overlapping area: The authors defined the open region and the coastal region as "covering broadly the open ocean at a distance of 1°. off the coast and, the second dataset, by Laruelle et al. (2017), covering the coastal domain plus the adjacent open ocean up until 400km away from the shoreline". And in page 6 line 2 the authors stated "landward limit of the NNopen is located on average at around 1° (or roughly 100km) offshore". As the authors know 1 degree latitude is almost 110.6 km to 111.7 km but 1 degree longitude depends on latitude and varied from 111.2 km to zero. Therefore the authors should make clear how they define and treat the data as the open ocean.**

*Response: We concur that using ° and km interchangeably without further explanation may cause confusion. The open ocean product is defined as the ocean area 1° away from shore, which is – as stated by the referee depending on geographical position – variable in km. The Laruelle estimate on the other hand uses the 400km definition, i.e. it is not variable depending on latitude. We have clarified this in the text at the positions indicated by the referee.*

*In particular, on page 6 lin2 we added: "While the landward limit of the NNopen is located at 1° (and therefore varies in km depending on the geographical position) off shore, …"*

*In the conclusions section we further added: " … leading to an overlap domain of roughly 300km close to the equator and increasing in extend towards the poles around the land surface"*

**R#2: 2, page 7. Figure 3 is important to understand how the authors merged the open ocean product and the coastal region product. Therefore it might better to enlarge this figure 3. The reviewer also suggests adding a numerical table to show an example of how they merged.**

*Response: We have now rearranged figure 3 so it appears larger in the manuscript (see figure (a) below). Additionally, we have added another figure (instead of a table – see (b) below) highlighting the statistics of the merging algorithm (new figure illustrated below including number of observations, mean differences and std differences within each 30x30 box). We believe that the newly introduced box-whisker plot is easier to grasp than an example highlighted in a numeric table.*

[Figure]

**R#2: 3, page 10 In the Figure 5, the maximum of a color bar of mismatch percent means that clear red indicates exceed 10 %. The reviewer suggests extending this color bar at least 15 % or 20 % to clearly show the regions where the mismatch is large because a smaller mismatch region does not need to highlight but a larger mismatch region should be highlighted.**

*Response: We have now increased the maximum value of the colorbar accordingly to 15% and changed the color palette to better highlight regions with larger mismatch (see updated figure below). We concur that we could further expand the upper limit, however, we would therefore miss to represent the geographical finer scale differences (e.g. along the Antarctic continent).*

[Figure]

**R#2: 4, Page 14 line 3. The authors discussed about Sea of Japan. It is however this region is a marginal sea and it not appropriate to compare NNopen and NNcoast here because the Sea of Japan might be included into coastal region following 400 km definition from the Japanese coast and Korean/Russian coast. Furthermore, there are probably no observed data at the Korean/Russian side based on Figure 9 ( c ). Therefore it is better to delete this part from this article.**

*Response: Many thanks for this keen observation. As can been seen in Figure 2 and Figure 9 of the manuscript, both open ocean and coastal ocean datasets in the SOCAT databases include measurements from the Sea of Japan. That said, we believe that including a marginal Sea in this intercomparison is an exciting opportunity to compare how both open ocean and coastal ocean reconstructions are able to represent in a marginal sea. We see this as relevant information to users who want to use the product to investigate this and other marginal seas. As illustrated in Figure 9 e and f, both products struggle to reproduce the available data, which indeed may be related to the fact that coastal and open ocean products have difficulties reconstructing the dynamics of this marginal Sea. So instead of removing this part, we have expanded the discussion of the mismatch in light of the fact that this region comprises a marginal sea.*

*In particular, we added in the first paragraph of the Regional Analysis section: ", two data rich regions (Sea of Japan, US east coast) of which one comprises a marginal sea (Sea of Japan), one*

*region where seasonal data are scarce (West Coast of Australia), and a region characterized by strong river outflow (Amazon river plume)."*

*We also extended the discussion regarding the Sea of Japan which now reads: "The strong variability in the observed pCO2 reflects the complex carbon dynamics in the Sea of Japan (Chen et al 1995, Park et al 2006), which is also reflected in the larger mismatch between products and towards the SOCAT observations (figures 10 d-f). The disagreement may indicate that the global scale NNopen and NNcoast products are not particularly skilled in representing the strong regional dynamics of marginal sea."*

*Finally, we added to the conclusions: "However, stronger differences exist in other parts of the world, particularly in the Peruvian upwelling system, the Arctic and Antarctic, the African coastline in the South Atlantic and the Arabian Sea, where fewer observations exist. Additionally, we find larger discrepancies in the marginal Sea of Japan."*

**R#2: 5, Figure 6,7,8,9,10,11,12: In (d)(e)(f) of these 7 figures, it is a little bit difficult to see the differences. Especially to distinguish difference zero region and no data region because the authors assigned no fill to both regions. Please re-draw these figures.**

*Response: We concur that differences close to 0 are more difficult to spot, and we have therefore adjusted the colorbar accordingly so that 0 values are not displayed white. We nevertheless chose a "soft color", i.e. yellow, to display values close to 0 as we intend to highlight discrepancies from 0 in these plots. Below is an example of the reworked figures (using the Amazon outflow as example region)*

[Figure]

**R#2: 6, P21 line 19- The authors stated that "Despite the lack of seasonal observations along the West coast of Australia, both products agree well with regards to the seasonal cycle and differences stay within of 8-10µatm between the different products.". The reviewer observed in figure 13 that in these three regions NNopen and NNcosat products showed a minimum or a maximum although there are no observed data at the time of a minimum or a maximum¸eg. a minimum in September on the west coast of Australia. The reviewer cannot understand how NNopen and NNcosat products there were produced and showed a minimum/maximum. Please explain this.**

*Response: Both products (coast and open ocean) are the result of a neural network interpolation of all available observations regressed onto driver data (see also methods here and in Landschützer et al 2014 and Laruelle et al 2017 cited in this work). Whenever there are no local observations available, the pCO2 is reconstructed from observations that fall within the same biogeochemical province, defined by a self-organizing map algorithm. In a second step all observations from the same province are regressed against physical (temperature, salinity, mixed layer depth), chemical (atmospheric CO2) and biological (chlorophyll a) driver data using a non-linear neural network-based regression approach (a feed-forward network). Based on the variability of these driver data the resulting pCO2 fields show variability in space and time and – in this particular case – a minimum in September largely owing (as we believe) to the solubility pump.*

**R#2: 7, Page 21 line 17- The authors stated that "Therefore, the combined pCO2 climatology is not only a step forward in including the  full oceanic domain with all its complexity into carbon budget analyses, but also help identify areas where additional continuous observations are critically needed to close current knowledge gaps.". The reviewer completely agree this statement and would like to suggest to add some recommendations explicitly from the authors to the community about areas where additional continuous observations are critically needed to close current knowledge gaps. If the authors do so, the contribution of this article to the community will increase much.**

*Response: We now expanded on this statement to provide explicit recommendations based on the findings of this manuscript. In particular, we mentioned the Peru upwelling system and high latitudes as prime example, since we face a critical monthly difference between open ocean and coastal ocean reconstructions, and we believe that this huge gap cannot be closed improving the methods, but only by observing the true pCO2.*

*In particular, we added to the conclusions: "The overlap analysis proposed here and particularly the Percent mismatch and RMSE analysis, further serves as a benchmark on how well we understand the coastal-to-open ocean continuum and its spatial variability and where we still lack essential measurements to close the gap between existing estimates, such as e.g. the Peruvian upwelling system or the seasonally ice-covered high latitude regions, in particular the Arctic Ocean"*